# Dataset search in biodiversity research: Do metadata in data repositories reflect scholarly information needs?

**Felicitas Löffler** [ID][1]*, **Valentin Wesp** [ID][1], **Birgitta König-Ries** [ID][1,2,3], **Friederike Klan** [ID][2,4]

**1** Heinz Nixdorf Chair for Distributed Information Systems, Department of Mathematics and Computer Science, Friedrich Schiller University Jena, Jena, Germany, **2** Michael-Stifel-Center for Data-Driven and Simulation Science, Jena, Germany, **3** German Center for Integrative Biodiversity Research (iDiv), Halle-Jena-Leipzig, Germany, **4** Citizen Science Group, DLR-Institute of Data Science, German Aerospace Center, Jena, Germany

* felicitas.loeffler@uni-jena.de

**Data Availability Statement:** The code and data are available in GitHub repository: https://github.com/fusion-jena/QuestionsMetadataBiodiv. In addition, the data has been submitted to the iDiv data portal (https://idata.idiv.de/). https://doi.org/

## Abstract

The increasing amount of publicly available research data provides the opportunity to link and integrate data in order to create and prove novel hypotheses, to repeat experiments or to compare recent data to data collected at a different time or place. However, recent studies have shown that retrieving relevant data for data reuse is a time-consuming task in daily research practice. In this study, we explore what hampers dataset retrieval in biodiversity research, a field that produces a large amount of heterogeneous data. In particular, we focus on scholarly search interests and metadata, the primary source of data in a dataset retrieval system. We show that existing metadata currently poorly reflect information needs and therefore are the biggest obstacle in retrieving relevant data. Our findings indicate that for data seekers in the biodiversity domain environments, materials and chemicals, species, biological and chemical processes, locations, data parameters and data types are important information categories. These interests are well covered in metadata elements of domain-specific standards. However, instead of utilizing these standards, large data repositories tend to use metadata standards with domain-independent metadata fields that cover search interests only to some extent. A second problem are arbitrary keywords utilized in descriptive fields such as title, description or subject. Keywords support scholars in a full text search only if the provided terms syntactically match or their semantic relationship to terms used in a user query is known.

## Introduction

Scientific progress in biodiversity research, a field dealing with the diversity of life on earth—the variety of species, genetic diversity, diversity of functions, interactions and ecosystems [1] -, is increasingly achieved by the integration and analysis of heterogeneous datasets [2, 3]. Therefore, locating and finding proper data for synthesis is a key challenge in daily research practice. Datasets can differ in format and size. Interesting data is often scattered across

10.25829/idiv.1864-15-2987 (Question Corpus)
https://doi.org/10.25829/idiv.1881-12-3202
(Metadata Analysis).

**Funding:** We acknowledge the Collaborative Research Centre AquaDiva (CRC 1076 AquaDiva, DFG Project Number: 218627073) of the Friedrich Schiller University Jena, the GFBio project (DFG Project Number: 229241684) and the Open Access Publication Fund of the Thueringer Universitaets- und Landesbibliothek Jena (DFG Project Number: 433052568), all funded by the Deutsche Forschungsgemeinschaft (DFG). The funder had no role in study design, data collection and analysis, decision to publish, or preparation of the manuscript.

**Competing interests:** The authors have declared that no competing interests exist.

various repositories focusing on different domains. In a survey conducted by the Research Data Alliance (RDA) Data Discovery Paradigm Interest Group [4], 35% of the 98 participating repositories stated that they host data from Life Science and 34% indicated they cover Earth Science. All of these are potentially of interest to biodiversity researchers.

However, the offered search services at public data providers do not seem to support scholars effectively. A study by Kacprzak et al. [5] reports that 40% of users attempting to search within two open data portals could not find the data they were interested in, and thus directly requested the data from the repository manager. In several studies, ecologists report on the difficulties they had when looking for suitable datasets to reuse [3, 6, 7]. In a recent, large-scale survey by Gregory et al [8], the vast majority of participants characterized data discovery as a sometimes challenging (73%) or even difficult (19%) task. 49% of the respondents stated that the greatest obstacles are "inadequate search tools, a lack of skill in searching for data, or the fact that their needed data are not digital" [8]. Hence, many researchers do not search for data in repositories, but rather follow data mentions in literature or employ general search engines for their dataset search [8]. Thus, there is a high demand for new techniques and methods to better support scholars in finding relevant data in data repositories.

In this study, we explore what hampers dataset retrieval in data repositories for biodiversity research. We analyze two building blocks in retrieval systems: *information needs (user queries)* and underlying *data*. We want to find out how large the gap is between scholarly search interests and provided data. In order to identify scholarly search interests, we analyzed *user questions*. In contrast to user queries, which are usually formulated in a few keywords, questions represent a search context, a more comprehensive information need. Characteristic terms or phrases in these textual resources can be labeled and classified to identify biological entities [9, 10]. *Scientific data* are not easily accessible by classical text retrieval mechanisms as they were mainly developed for unstructured textual resources. Thus, effective data retrieval heavily relies on the availability of proper *metadata* (structured information about the data) describing available datasets in a way that enables their *Findability*, one principle of FAIR data [11]. A survey conducted by the Research Data Alliance (RDA) Data Discovery Group points out that 58% of the 98 participating data repositories index all metadata and partial metadata (52%). Only one third of the participating repositories integrate data dictionaries or variables (the actual primary data) [4].

We argue that *Findability* at least partially depends on how well metadata reflect scholarly information needs. Current retrieval evaluation methods are basically focused on improving retrieval algorithms and ranking [12, 13]. Therefore, question corpora and documents are taken as given and are not questioned. However, if the underlying data do not contain the information users are looking for, the best retrieval algorithm will fail. We argue that in dataset search, metadata, the basic source for dataset applications, need to be adapted to match users' information needs. Thus, the following analysis aims to explore:

- What are genuine user interests in biodiversity research?

- Do existing metadata standards reflect information needs of biodiversity scientists?

- Are metadata standards utilized by data repositories useful for data discovery? How many metadata fields are filled?

- Do common metadata fields contain useful information?

We propose a top-down approach starting from scholars' search interests, then looking at metadata standards and finally inspecting the metadata provided in selected data repositories:

(A) We first identified main entity types (categories) that are important in biodiversity research. We collected 169 questions provided by 73 scholars of three large and very diverse biodiversity projects in Germany, namely *AquaDiva* [14], *GFBio—The German Federation for Biological Data* [15] and *iDiv—The German Research Center for Integrative Biodiversity Research* [1]. Two authors of this publication labeled and grouped all noun entities into 13 categories (entity types), which were identified in several discussion rounds. Finally, all proposed categories were evaluated with biodiversity scholars in an online survey. The scholars assigned the proposed categories to important phrases and terms in the questions (Section "A—Information Needs in the Biodiversity Domain").

(B) Most data providers use keyword-based search engines returning data sets that exactly match keywords entered by a user [4]. In dataset search, the search index mainly consists of metadata. The metadata schema required or recommended by the data repository greatly influences the richness of metadata descriptions and determines available facets for filtering. Therefore, we inspected common metadata standards in the Life Sciences and analyzed, to which extent their metadata schemata cover the identified information categories (Section "B—Metadata Standards in the Life Sciences").

(C) There are several data repositories that take and archive scientific data for biodiversity research. According to *Nature's* list of recommended data repositories [16], repositories such as *Dryad* [17], *Zenodo* [18] or *Figshare* [19] are generalist repositories and can handle different types of data. Data repositories such as *PANGAEA* [20] (environmental data) or *GBIF* [21] (occurrence data) are domain specific and only take data of a specific format. We harvested and parsed all publicly available metadata from these repositories and determined whether they utilize metadata elements that reflect search interests. For *GBIF*, we concentrated on datasets only, as individual occurrence records are not available in the metadata API. We explored how many fields of the respective schemata are actually used and filled (Section "C—Metadata Usage in Selected Data Repositories").

(D) Finally, we discuss the results and outline how to consider and address user interests in metadata (Section "D—Discussion").

In order to foster reproducibility, questions, scripts, results and the parsed metadata are publicly available: https://github.com/fusion-jena/QuestionsMetadataBiodiv.

The structure of the paper is as follows: The first part "Definitions" focuses on the clarification of various terms. Afterwards, we introduce "Related Work". The following four sections contain the individual research contributions described above. Each of these sections describes the respective methodology and results. Finally, section "Conclusion" summarizes our findings.

## Definitions

Since dataset retrieval is a yet largely unexplored research field [22], few definitions exist describing what it comprises and how it can be characterized. Here, we briefly introduce an existing definition and add our own definition from the Life Sciences' perspective.

Chapman et al [22] define a dataset as "A collection of related observations organized and formatted for a particular purpose". They further characterize a dataset search as an application that "involves the discovery, exploration, and return of datasets to an end user." They distinguish between two types: (a) a basic search in order to retrieve individual datasets in data portals and (b) a constructive search where scholars create a new dataset out of various input datasets in order to analyze relationships and different influences for a specific purpose.

From our perspective, this definition of a dataset is a bit too restrictive. All kinds of scientific data such as experimental data, observations, environmental and genome data, simulations and computations can be considered as datasets. We therefore extend the definition of Chapman et al [22] as follows:

**Definition 1**. *A **dataset** is a collection of scientific data including primary data and metadata organized and formatted for a particular purpose.*

We agree with Chapman et al.'s definition of dataset search. We use *dataset search* and *dataset retrieval* synonymously and define it as follows:

**Definition 2**. ***Dataset retrieval** comprises the search process, the ranking and the return of scientific datasets.*

Unger et al. [23] introduced three dimensions to take into account in question answering namely the *User* and *Data* perspective as well as the *Complexity* of a task. We argue that these dimensions can also be applied in dataset retrieval. In this work, we do not further explore the complexity of questions. We focus on the analysis of user interests and metadata, only. However, for completeness and possible future research in computer science towards question answering in dataset retrieval, the analysis of the complexity of search questions and queries is essential. Therefore, we briefly introduce all three dimensions.

## User perspective

In conventional retrieval systems users' search interests are represented as a few keywords that are sent to the system as a search query. Keywords are usually embedded in a search context that can be expressed in a full sentence or a question.

In order to understand what users are looking for, a semantic analysis is needed. *Information Extraction* is a technique from text mining that identifies main topics (also called entity types) occurring in unstructured text [24]. Noun entities are extracted and categorized based on rules. Common, domain-independent entity types are for instance Person, Location and Time. When it comes to specific domains, additional entity types corresponding to core user interests need to be taken into consideration. In bio-medicine, according to [25], the main topics are Data Type, Disease Type, Biological Process and Organism. In new research fields such as biodiversity research these main entity types still need to be identified in order to get insights into users' information needs and to be able to later adapt systems to user requirements.

## Data perspective

From the data perspective, a dataset search can be classified into two types based on the source of data: *primary data* and *metadata*.

**Definition 3**. ***Primary data** are scientific raw data. They are the result of scientific experiments, observations, or simulations and vary in type, format, and size.*

**Definition 4**. ***Metadata** are structured, descriptive information of primary data and answer the W-questions: **What** has been measured by **Whom**, **When**, **Where** and **Why**?. Metadata are created for different purposes such as search, classification, or knowledge derivation.*

Dataset retrieval approaches focusing on primary data as source data have to deal with different data formats such as tabular data, images, sound files, or genome data. This requires specific query languages such as QUIS [26] to overcome the ensuing heterogeneity, which is out of scope of this paper. Here, we solely focus on dataset retrieval approaches that use metadata as input for search. A variety of metadata standards in the Life Sciences are introduced in Section "B—Metadata Standards in the Life Sciences".

## Complexity

Scholarly search interests are as heterogeneous as scientific data. Concerning complexity, information needs can either be very specific and focus on one research domain only, or search interests can be broader and comprise various research fields. In terms of the search results, retrieved datasets may contain the complete answer to a search query or answer queries only partially. For the latter, users might construct new datasets out of various input datasets. Unger et al. [23] characterize the complexity in retrieval tasks along four dimensions: *Semantic complexity* describes how complex, vague, and ambiguous a question is formulated and if heterogeneous data have to be retrieved. *Answer locality* denotes if the answer is completely contained in one dataset or if parts of various datasets need to be composed or if no data can be found to answer the question. *Derivability* describes if the answer contains explicit or implicit information. The same applies for the question. If broad or vague terms appear in the question or answer, additional sources have to be integrated to enrich question and/or answer. *Semantic tractability* denotes if the natural language question can be transformed into a formal query.

## Related work

This section focuses on approaches that analyze, characterize and enhance dataset search. We discuss studies identifying users' information needs and introduce existing question corpora used for retrieval evaluation. In a second part, we describe what retrieval methods are used in existing data portals, and we introduce approaches that aim at enhancing dataset search.

## User interests

Query logs, surveys or question corpora are valid sources for studying user behavior in search. Kacprazal et al [5] provide a comprehensive log analysis of three government open data portals from the United Kingdom (UK), Canada, and Australia and one open data portal with national statistics from the UK. 2.2 million queries from logs provided by the data portals (internal queries) and 1.1 million queries issued to external web search engines (external queries) were analyzed. Two authors manually inspected a sample set of 665 questions and determined the main query topics. Most queries were assigned to Business and Economy (20% internal queries, 10% external queries) and Society (14.7% internal queries, 18% external queries). Besides query logs, Kacprazal et al [5] also analyzed explicit requests by users for data via a form on the website. Here, users provided title and description which allowed the authors to perform a deeper thematic analysis on 200 manually selected data requests. It revealed that geospatial (77.5%) and temporal (44%) information occurred most, often together with a specific granularity (24.5%), e.g., "hourly weather and solar data set" or "prescription data per hospital". Users were also asked why they had requested data explicitly, and more than 40% indicated that they were not able to find relevant data via the provided search.

A large study on user needs in biodiversity research was conducted in the *GBIF* community in 2009 [27, 28]. The aim was to determine what *GBIF* users need in terms of primary data and to identify data gaps in the current data landscape at that time. More than 700 participants from 77 countries took part in the survey. It revealed that scholars retrieved and used primary data for analyzing species diversity, taxonomy, and life histories/ phenology. That mainly required "taxon names, occurrence data and descriptive data about the species" [28]. As biodiversity is a rapidly changing research field, the authors recommended to repeat content need assessments in frequent intervals [27].

Apart from query logs and surveys, question corpora are another source for identifying search interests. Usually, questions are collected from experts of a particular research field and important terms representing main information needs are labeled with categories or so-called

*entity types*. Manually generated annotations are helpful in understanding what information users are interested in and support the development of tools and services to either automatically extract these interests from text (text mining), to retrieve relevant data (information retrieval) or to provide an exact answer for that information need (question answering).

In the Life Sciences, question corpora for text retrieval have been mainly established in the medical and bio-medical domains. Genuine user requests have been collected in surveys for the Genomics Track at TREC conferences [29] and in email requests (67%) and search query logs (33%) for the Consumer Health Corpus [9]. In contrast, the question corpora for the BioASQ challenge [10], an annual challenge for researchers working on text mining, machine learning, information retrieval and question answering, was created and annotated by a team of 10 experts [30] with concrete requirements. Each expert was asked to formulate 50 questions in English that reflect "real-life information needs". However, the type of questions to be formulated was restricted, e.g., the experts were asked to provide questions of certain types typically considered in question answering systems (yes/no, factoid, etc.). These restrictions are justified to a certain degree since they affect the applicability of the resulting corpus for evaluation purposes of question answering approaches. However, they have an impact on which questions are formulated and how. A question corpus dedicated for dataset search retrieval is included in the benchmark developed for the 2016 bioCADDIE Dataset Retrieval Challenge [25]. It includes 137 questions, 794,992 datasets gathered from different data portals in XML structure, and relevance judgments for 15 questions. Similar to the BioASQ challenge, domain experts had to consider guidelines on how to create questions. Based on templates, the question constructors formulated questions using the most desired entity types, namely data type, disease type, biological process, and organism.

At present, to the best of our knowledge, there is neither a public log analysis nor a question corpus available for biodiversity research. In order to understand genuine user interests and to improve current dataset retrieval systems, unfiltered information needs are crucial. Therefore, collecting current search interests from scholars is the first step in our top-down approach presented in Section "A—Information Needs in the Biodiversity Domain".

## Dataset search

Dataset search is mainly based on metadata describing heterogeneous primary data. Terminologies used in metadata are influenced by scholars' research fields. As data in the Life Sciences are difficult to classify and categorize for humans [31, 32], it is an even larger challenge to develop a system that can interpret user queries correctly and return relevant datasets.

The main aim of a retrieval process is to return a ranked list of datasets that match a user's query. Based on the underlying *Retrieval Model*, different ranking functions have been developed to produce a score for the datasets with respect to the query. Top-scored datasets are returned first. Classical retrieval models include the *Boolean Model* [12], where datasets are only returned that exactly match a query, and the *Vector Space Model* [12], often used in combination with the Boolean Model. In the Vector Space Model, the content of datasets is represented by vectors that consist of term weights. The similarity of datasets and queries is determined by computing the distance between the vectors.

All these retrieval methods have in common that they only return datasets that exactly match a user's query. A study by the RDA Data Discovery Paradigm Interest Group points out [4] that most data repositories utilize either *Apache Solr* (http://lucene.apache.org/solr/) or *elasticsearch* (https://www.elastic.co/products/elasticsearch). The default retrieval method in these search engines is either one of the mentioned keyword-based retrieval models or a combination of both. *Apache Solr* and *elasticsearch* provide fuzzy-search

mechanisms to handle misspellings but full semantic search approaches are very rare in large data repositories (see also Subsection "Challenges to be addressed in Computer Science"). Hence, if the desired information need is not explicitly mentioned in the metadata, the search will fail. This is reinforced by the fact that some repositories only integrate partial metadata in their search index [4]. In order to provide improved user interfaces supporting exploratory search paradigms [33], data repositories increasingly offer faceted search facilities. Facets are a group of meaningful labels comprising relevant categories of a domain [34]. A faceted search allows users to influence the result set and to narrow its scope, e.g., to filter for datasets in a specific geographic region. If not manually created, facets need to be built on metadata fields or grouped keywords. Numerous studies in computer science explore automatic approaches for facet creation [35–37]. Recent studies also consider the usage of semantic knowledge bases (e.g., [38, 39]). However, so far these techniques have not been integrated into existing search systems in dataset retrieval and can not completely replace predefined domain categories.

In recent years, a variety of approaches have emerged to improve dataset search. A common approach is to annotate metadata with entities from *schema.org* (https://schema.org). Favored by Google [40] and the RDA Data Discovery Paradigm Interest Group [41], the idea is to add descriptive information to structured data such as XML or HTML in order to increase findability and interoperability. These additional attributes help search engines to better disambiguate terms occurring in text. For example, Jaguar could be a car, an animal or an operating system. By means of *schema.org* entities, data providers can define the context explicitly. Numerous extensions for specific domains have been developed or are still in development, e.g., *bioschemas.org* [42] for the Life Sciences. Since Google launched its beta version of a dataset search in Fall 2018 (https://toolbox.google.com/datasetsearch), *schema.org* entities got more and more attention. Hence, data centers such as *PANGAEA* [20] or *Figshare* [19] are increasingly incorporating *schema.org* entities in their dataset search.

Other approaches favor an improved metadata schema. Pfaff et al [43] introduce the Essential Annotation Schema for Ecology (EASE). The schema was primarily developed in workshops and intensive discussions with scholars and aims to support scientists in search tasks. EASE consists of eight top level categories (Time, Space, Sphere, Biome, Organism, Process, Method and Chemical) that were enriched with further elements in a top down approach. Finally, EASE consists of around 1600 elements. The MIBBI project [44] also recognized that only improved metadata allow information seekers to retrieve relevant experimental data. They propose a harmonization of minimum information checklists in order to facilitate data reuse and to enhance data discovery across different domains. Checklist developers are advised to consider "'cross-domain' integrative activities" [44] when creating and maintaining checklists. In addition, standards are supposed to contain information on formats (syntax), vocabularies and ontologies used. Nowadays, its successor *FAIRsharing* [45] manually curates metadata on these standards and the relationships between them and relates this back to metadata on the repositories and knowledge bases that implement and use them. Further, *FAIRsharing* links both standards and databases to journal and funder data policies that recommend or endorse their use.

Federated approaches attempt to align heterogeneous data sources in one search index. That allows the use of conventional search engines and keyword-based user interfaces: DataONE is a project aiming to provide access to earth and environmental data provided by multiple member repositories [46]. Participating groups can provide data in different metadata formats such as EML, DataCite or FGDC [47]. DataONE is currently working on quantifying FAIR [48]. Their findability check determines if specific metadata items such as title, abstract or publication date are present. For title and abstract, they additionally check the length and

content. Based on these criteria, they evaluated their data and found out that concerning *Findability* around 75% of the available metadata fulfilled the self-created criteria. The German Federation for Biological Data (GFBio) [49] is a national infrastructure for research data management in the green Life Sciences and provides a search over more than six million heterogeneous datasets from environmental archives and collection data centers.

As described above, numerous approaches have been proposed and developed to improve dataset search. However, what is lacking is a comprehensive analysis on what exactly needs to be improved and how large the actual gap is between user requirements and given metadata.

## A—Information needs in the biodiversity domain

Question corpora are common sources to determine user interests for a particular domain. Therefore, we asked biodiversity scholars to provide questions that are specific for their research. We analyzed the questions and identified search topics that represent scholarly information needs in this domain.

### Methodology

Our methodology for the question analysis is based on common question collection and annotation procedures in information retrieval and question answering [9, 10, 50]. In the following subsection, we describe the methodology in detail. It comprises four parts: (1) question collection, (2) category definition, (3) annotation process, (4) annotators and category assignment.

**(1) Question collection.** We gathered questions in three large biodiversity projects, namely *CRC AquaDiva* [14], *GFBio* [15] and *iDiv* [1]. We explicitly requested fully expressed questions to capture the keywords in their search context. These projects vary widely in their overall setting, the scientists and disciplines involved and their main research focus. Together, they provide a good and rather broad sample of current biodiversity research topics. In total, 73 scholars with various research backgrounds in biology (e.g., ecology, bio-geochemistry, zoology and botany) and related fields (e.g., hydro-geology) provided 184 questions. This number is comparable to related question corpora in Information Retrieval (e.g., bioCADDIE [25]) which typically consist of around 100—150 questions. The scholars were asked to provide up to five questions from their research background. Questions varied with respect to granularity. The corpus contains specific questions, such as *List all datasets with organisms in water samples.* as well as questions with a broader scope, e.g., *Does agriculture influence the groundwater?*. We published the questionnaires that were handed out in *AquaDiva* and *iDiv* as supplementary material in our repository. In the *GFBio* project, questions were gathered via email and from an internal search evaluation. All questions were inspected by the authors with respect to comprehensibility. We discarded questions that were not fully understandable (e.g., missing verb, misleading grammatical structures) but left clear phrases in the corpus that were not fully expressed as a question. If scholars provided several questions, they were treated individually even if terms referred to previous questions, e.g., *Do have earthworm burrows (biopores) an impact on infiltration and transport processes during rainfall events?* and *Are the surface properties influencing those processes?*. In this case, no further adaption towards comprehensibility has been made. The questions were also not corrected with respect to grammar and spelling since changing the grammar could lead to an altered statement. We did not want to lose the original question statement. In some questions, abbreviations occurred without explanations. In these cases, we left the questions as they are and did not provide full terms, since these abbreviations can have various meanings in different biological fields. It was up to the domain experts to either look them up or to leave the term out. After the cleaning, the final

corpus consists of 169 questions and is publicly available: https://github.com/fusion-jena/QuestionsMetadataBiodiv/tree/master/questions.

(2) **Category definition.** Boundaries of semantic categories are domain-dependent and fuzzy. However, in search, categories support users in finding relevant information more easily and should be valid across various research backgrounds. In a first round, two authors of this work analyzed the collected questions manually. Both have a research background in computer science and strong knowledge in scientific data management, in particular for biodiversity research. The corpus was split up and each of them inspected around 50% of it and assigned broad categories independently. Afterwards, this first classification was discussed in several sessions. This resulted in 13 categories. The naming was adapted to domain-specific denotations and ontologies. Furthermore, the categories were compared to EASE [43], a metadata schema which was primarily developed for an improved dataset retrieval in the field of ecology. This comparison revealed that there is an overlap with EASE but that we discovered further relevant categories [51]. The final categories are:

1. ORGANISM comprises all individual life forms including plants, fungi, bacteria, animals and microorganisms.

2. All species live in certain local and global ENVIRONMENTS such as habitats, ecosystems (e.g., below 4000 m, ground water, city).

3. Species and environments have certain characteristics (traits, phenotypes) that can be observed, measured or computed. These data parameters are summarized with QUALITY & PHENOTYPE, e.g., length, growth rate, reproduction rate.

4. Biological, chemical and physical PROCESSES are re-occurring and transform materials or organisms due to chemical reactions or other influencing factors.

5. EVENTS are processes that appear only once at a specific time, such as environmental disasters, e.g., Deepwater Horizon oil spill, Tree of the Year 2016.

6. Chemical compounds, rocks, sand and sediments can be grouped as MATERIALS & SUBSTANCES.

7. ANATOMY comprises the structure of organisms, e.g., body or plant parts, organs, cells, and genes.

8. METHOD describes all operations and experiments that have to be conducted to lead to a certain result, e.g., lidar measurements, observation, remote sensing.

9. Outcomes of research methods are delivered in DATA TYPE, e.g., DNA data or sequence data is the result of genome sequencing, lidar data is the result of lidar measurements (active remote sensing).

10. All kinds of geographic information is summarized with LOCATION, e.g., Germany, Hainich National Park, Atlantic Ocean, and

11. temporal data including date, date times, and geological eras are described by TIME, e.g., current, over time, Triassic.

12. PERSON & ORGANIZATION are either projects or authors of data.

13. As reflected in the search questions, scholars in biodiversity are highly interested in HUMAN INTERVENTION on landscape and environment, e.g., fishery, agriculture, and land use.

For the evaluation with domain experts we added two more categories, namely OTHER and NONE. The first permits to define a new category, if none of the given ones is appropriate. If an artifact got no rating, we counted it with the category NONE.

**(3) Annotation process.**  An annotation process usually has two steps: the identification of terms based on annotation rules and the assignment of an appropriate category in a given context. Usually, an annotator, a domain expert, who is trained in the annotation guidelines, carries out both tasks. However, we argue that training is somewhat biased and influences annotators in their classification decision. This is an obstacle in search where an intuitive feedback for category assignment is required. Hence, we split up the annotation process. Two scholars, who collected the questions and who are familiar with the guidelines conducted the identification, whereas domain experts only received short instructions and assigned categories. Our annotation guidelines needed to identify phrases and terms (artifacts) to label, including definitions of the terminology used, are available as supplementary material in our GitHub repository (https://github.com/fusion-jena/QuestionsMetadataBiodiv). We assume, that this shortened process also reduces the annotation time for the annotators and might motivate more experts to complete this classification task.

**(4) Annotators and category assignment.**  Nine domain experts (8 Postdocs, 1 Project Manager) with expertise in various biological and environmental sciences participated in the classification task. All of them have experience in ecology but in addition, each of them has individual research competence in fields such as bio-geography, zoology, evolutionary biology, botany, medicine, physiology, or biochemistry.

For the category assignment, all scholars received a link to an online survey with explanations of the categories (including examples) and short instructions on how to classify the artifacts. A screenshot of the survey is presented in Fig 1. The purpose of this evaluation was also explained to them (improvement of data set retrieval systems). Multi-labeling was not allowed; only one category was permitted per artifact. The annotators were asked to assign the category that is most appropriate depending on the question context, e.g., water can occur in a chemical context, as food or as environment. But considering water in its given context usually limits the category selection. Therefore, in order to obtain the most intuitive feedback, we only allowed one category. Should there be no proper category, the annotators had the opportunity to select OTHER and if possible to provide an alternative category. If they did not know a term or phrase, they could decide either to look it up or to omit it. If they considered a phrase or

How important is *CO2 fixation* in the *groundwater*?

| | Organism | Environment | Quality | Mat & Subst | Process | Method | Data Type | Anatomy | Location | Time | Event | Person & Org | Human Inter | other |
|---|---|---|---|---|---|---|---|---|---|---|---|---|---|---|
| CO2 fixation | ○ | ○ | ○ | ○ | ● | ○ | ○ | ○ | ○ | ○ | ○ | ○ | ○ | |
| CO2 | ○ | ○ | ○ | ● | ○ | ○ | ○ | ○ | ○ | ○ | ○ | ○ | ○ | ○ |
| groundwater | ○ | ● | ○ | ○ | ○ | ○ | ○ | ○ | ○ | ○ | ○ | ○ | ○ | ○ |

**Fig 1. Excerpt of the survey that was set up for the classification task.** The annotators were told to assign only one category per given artifact. If an artifact is a compound noun, the nested entities such as adjectives or second nouns that further describe the term were provided for tagging as well. In this question, 'CO2 fixation' is an example for a two term artifact and 'groundwater' an example for a one term artifact.

term to be not relevant or too complicated and fuzzy, omission was also permitted. We counted all artifacts that did not get any rating with the category NONE.

As we wanted to obtain intuitive feedback, the experts were told not to spend too much time on the classification decision but to determine categories according to their knowledge and research perspective. For each question, annotators had the opportunity to provide a comment.

We decided to use a combination of csv files, Python scripts and *Limesurvey* to support the annotation process. Details on this process can be found in the supplementary material in our repository.

## Results and metrics

We analyzed the user responses to determine whether the identified information categories are comprehensive and representative for biodiversity research. We computed the inter-rater agreement per artifact to determine the category that best describes an artifact.

**Representativeness of the categories.** In order to verify completeness we determined the fraction of artifacts assigned to the category OTHER, i.e., if the experts deemed none of the given categories as appropriate. Fig 2 depicts the frequency of information categories and how often they were selected by the domain experts. As it turned out, the category OTHER was selected by at least 1 expert per artifact for 46% of the phrases and terms and by at least 2 experts for 24%. The fraction of phrases for which at least 3 experts selected the category

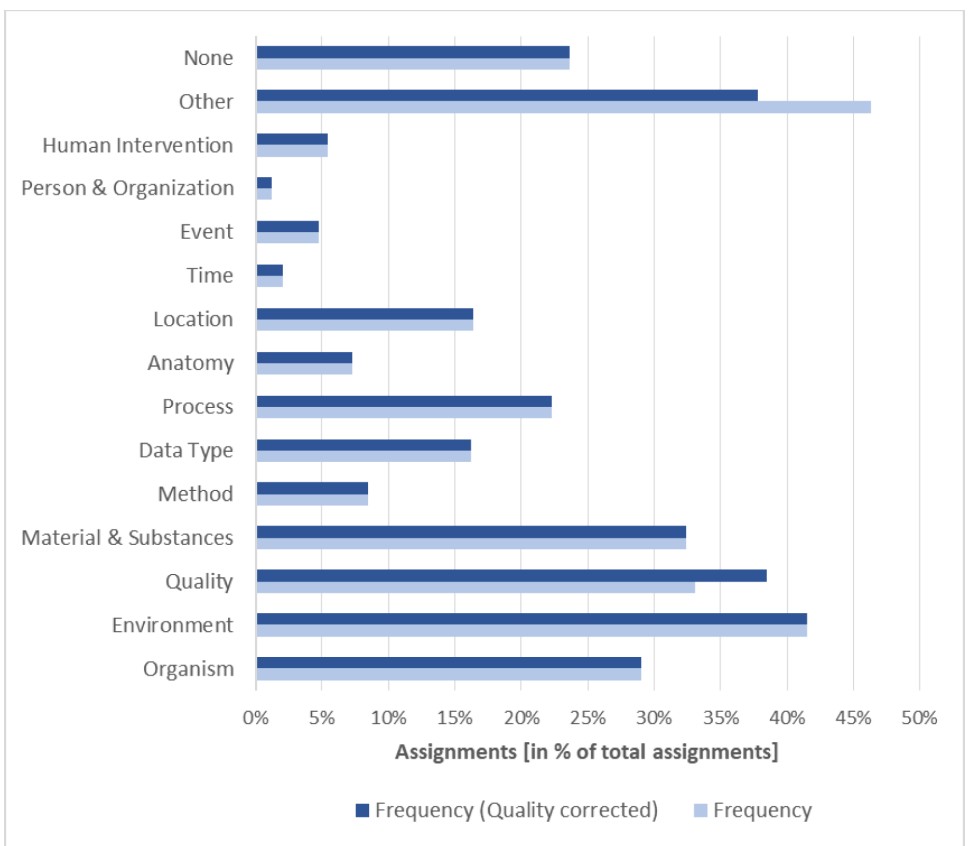

**Fig 2. The frequency of the categories and how often they were assigned to given phrases and terms, with and without QUALITY correction.**

OTHER was 12%. If at least two domain experts agree that there is no proper category for a given phrase, it is a strong indicator for a missing category or a misinterpretation. This is the case for almost a quarter out of all annotated artifacts. In other words, in around 75% of 592 annotations, 7 annotators assigned a given category. Hence, the coverage of the identified information categories is not complete, but still high.

However, there might be various reasons why none of the given categories fit: (1) The phrase or term to be annotated was unknown to the annotator such as *shed precipitation*. (2) Frequently, phrases that refer to data attributes (e.g., *soil moisture*, *oxygen uptake rate* or *amount of rain*) and which were supposed to be covered by the category QUALITY, were classified as OTHER. As alternative category, the annotators proposed "Parameter" or "Variable". When adding these ratings to the QUALITY category, the results for the OTHER category decreased to 37% (1 expert)/13% (2 experts)/4% (3 experts). That strongly indicates that renaming the QUALITY category or adding synonyms would increase comprehensibility significantly. (3) The category OTHER was often chosen for terms used in questions with a broader scope in order to express expected results. However, since this is often vague, scholars tend to use generic terms such as *signal*, *pattern*, *properties*, *structure*, *distribution*, *driver* or *diversity*. (4) A fourth reason could be the missing training. If the annotators did not understand a category, they might have been more likely to assign OTHER.

For all these terms in the category OTHER, further discussions in the biodiversity research community are needed to define and classify them.

In addition, we wanted to know if there are categories that were not or rarely used by the annotators. This would indicate a low relevance for biodiversity research. As depicted in Fig 2, the categories ENVIRONMENT, ORGANISM, MATERIAL & SUBSTANCES, QUALITY, PROCESS, LOCATION and DATA TYPE have been selected most frequently (assigned to more than 15% of the phrases). Information related to these categories seems to be essential for biodiversity research. Although there were categories that were rarely chosen (PERSON & ORGANISATION and TIME), there was no category that was not used at all.

**Consensus on the categories.** In statistics, the consensus describes how much homogeneity exists in ratings among domain experts. We determined the inter-rater agreement and inter-rater reliability using Fleiss' Kappa ($\kappa$ statistics) [52] and Gwet's AC [53]. In general, the inter-rater reliability computes the observed agreement among raters "and then adjusts the result by determining how much agreement could be expected from random chance" [54]. $\kappa$ values vary between −1 and + 1, where values less than 0 denote poorer than chance agreement and values greater than 0 denote better than chance agreement. As suggested by Landis and Koch [55], $\kappa$ values below 0.4 indicate fair agreement beyond chance, values between 0.4 and 0.6 moderate agreement, values between 0.6 and 0.8 substantial agreement and values higher than 0.80 indicate almost perfect agreement. However, $\kappa$ statistics can lead to a paradox: When the distribution of the raters' scores is unbalanced, the correction for the chance agreement can result in negative $\kappa$ values even if the observed agreement is very high [54]. Since this is the opposite of what is expected, a new and more robust statistic has emerged, the Gwet's AC [53]. Gwet's AC considers the response categories in the agreement by chance and the values can range from 0 to 1.

With a Fleiss' Kappa of 0.48 and Gwet's AC of 0.51 the agreement of the annotators over all categories was moderate. Considering the QUALITY correction, the values increase slightly to 0.49 for Fleiss' Kappa and 0.52 to Gwet's AC. Fig 3a reveals a more detailed picture. It shows the Fleiss' Kappa for the individual information categories with QUALITY correction. The agreement among the experts was excellent for the categories TIME and ORGANISM and intermediate to good for the categories PERSON & ORGANIZATION, LOCATION, PROCESS, MATERIALS & SUBSTANCES and ENVIRONMENT. The experts' agreement for the

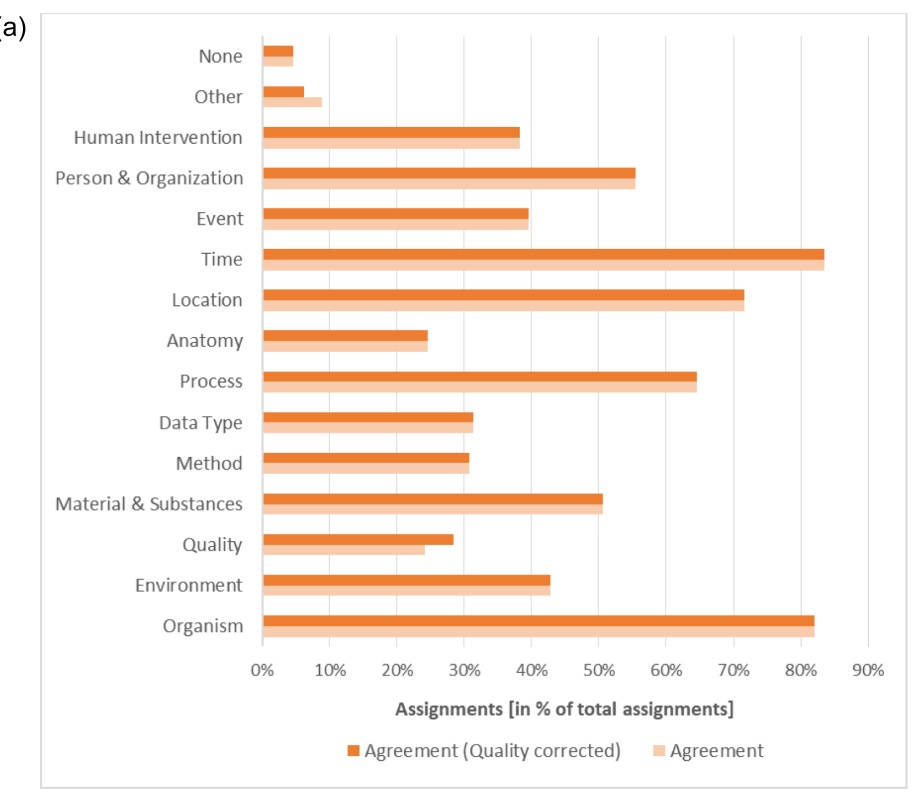

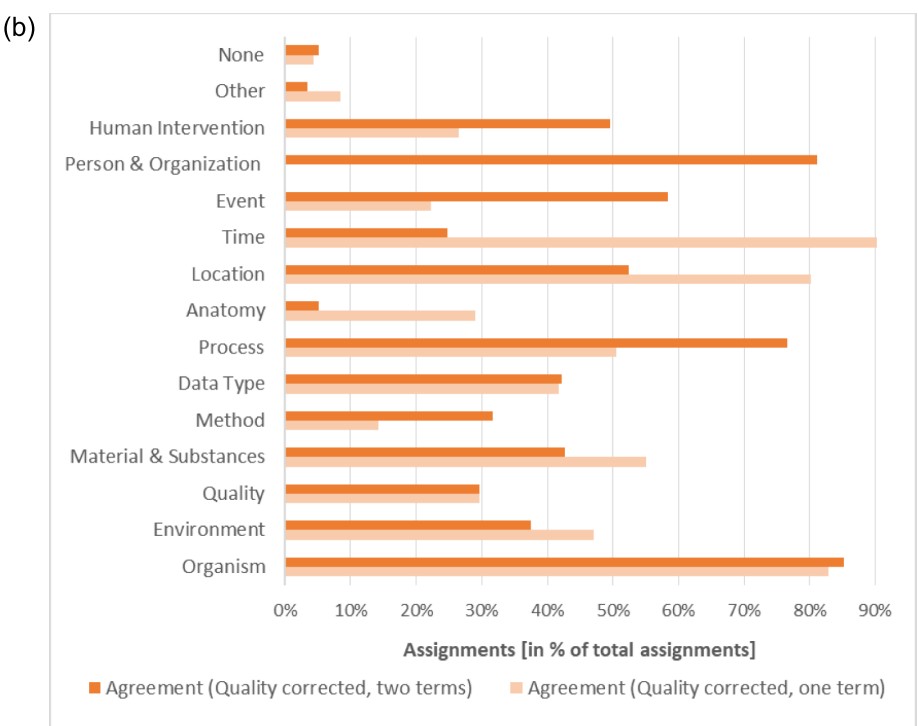

**Fig 3. Fleiss' Kappa values for the individual information categories (with QUALITY correction): a) for all artifacts b) for artifacts with one and two terms.**

**Table 1. Annotator's agreement with QUALITY correction overall and for one term, two terms, three terms and more per artifact.**

|  | Overall | One Term | Two Terms | > = Three Terms |
|---|---|---|---|---|
| *Fleiss' Kappa* | 0.49 | 0.54 | 0.50 | 0.33 |
| *Gwet' sAC* | 0.52 | 0.57 | 0.53 | 0.37 |

categories EVENT, HUMAN INTERVENTION, ANATOMY, DATA TYPE, METHOD and QUALITY was fair. This lack of agreement can either point to a different understanding of the categories or might indicate that the categorization of the phrase itself was difficult since some phrases, in particular longer ones with nested entities, were fuzzy and difficult to classify in one category. In the latter case, the annotators were advised not to choose a category for that phrase. Our results show that for 5% of the phrases at least 2 annotators did not provide a category. The fraction of phrases where 3 or more annotators did not choose a category was below 2%. A closer look at the specific results reveals that numerous entries in these cases contain comments pointing to a misunderstanding of the category QUALITY. For the categories EVENT, HUMAN INTERVENTION, ANATOMY, DATA TYPE, METHOD there is no further indication for the fair agreement. Here, further discussions with experts in biodiversity research are needed to analyze whether this is due to a category misunderstanding or whether other reasons have led to this result.

**Comparison of short and long artifacts.** We also analyzed the influence of longer artifacts on the result. Table 1 presents the $\kappa$ statistic and *Gwet' sAC* for artifacts with one term, two terms, three and more terms including the quality correction. As assumed, the longer an artifact is, the more difficult it is to assign an unambiguous category.

Fig 3b depicts a more detailed picture on the individual categories for artifacts with one and two terms. Since artifacts with three and more terms resulted in coefficients with less than 0.4, we left them out in this analysis. One-term artifacts got an excellent agreement (>0.8) for the categories ORGANISM, TIME and LOCATION and a moderate agreement for ENVIRONMENT, MATERIAL, PROCESS and DATA TYPE. It strikes that PERSON results in a negative value with a poor agreement. However, it is correct in this case and has a natural reason. Since full person names usually contain two terms, there were no artifacts with one term that could be assigned to PERSON & ORGANIZATION. Looking at the results for two terms per artifact, the PERSON category reaches an excellent agreement as well as ORGANISM. Surprisingly, PROCESS (0.76) got a substantial agreement for two terms pointing out that biological and chemical processes are obviously mainly defined by two terms. The same effect, a larger agreement for two terms than one term, can also be observed for the categories EVENT and HUMAN INTERVENTION. DATA TYPE got a moderate agreement for one and two terms.

**Summary.** All 13 provided categories were used by the annotators to label the artifacts in the questions. However, what stands out is the high number of the category OTHER in the frequency analysis. For 45% out of 592 annotations, at least one domain expert did not assign one of the given categories but selected OTHER. That points to missing interests that are not represented by the given classes.

In terms of consensus, seven information categories got a moderate agreement (> 0.4) and five out of these seven were also mentioned very often (>15%), namely ENVIRONMENT (e.g., habitats, climate zone, soil, weather conditions), MATERIAL (e.g., chemicals, geological information), ORGANISM (species, taxonomy), PROCESS (biological and chemical processes) and LOCATION (coordinates, altitude, geographic description) (Fig 4). We conclude that these classes are important search interests for biodiversity research.

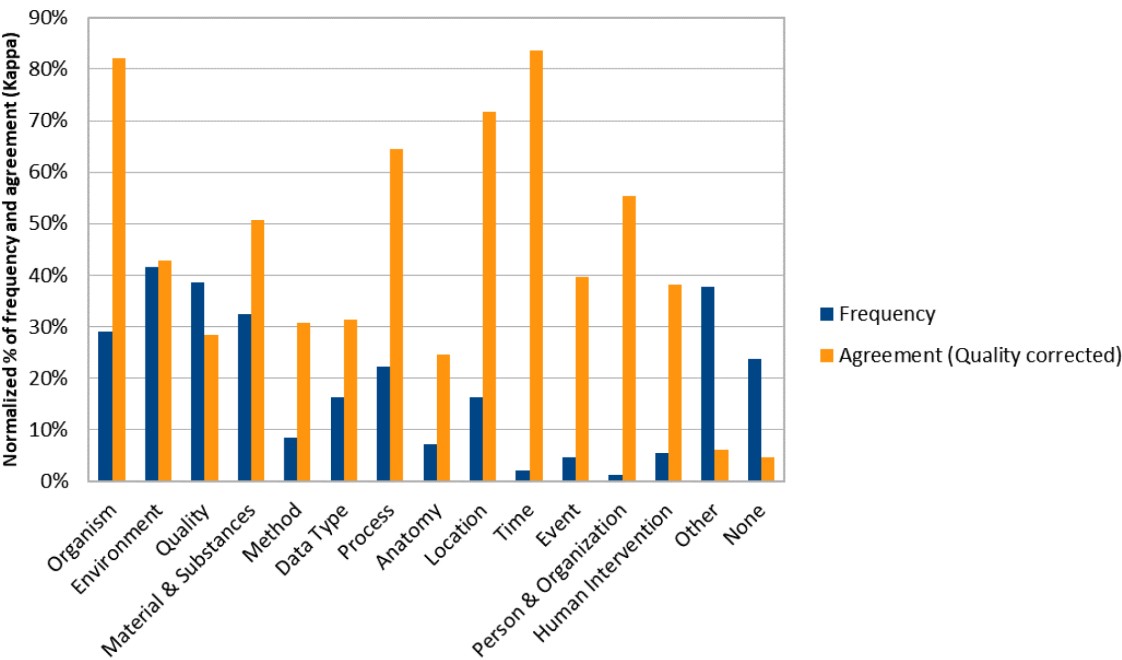

**Fig 4. Frequency of category mentions and inter-rater agreement with QUALITY correction.**

We are aware that this result is not complete and leaves room for improvement. Some category names were misleading and confused the annotators. That is reflected in fair and bad agreement for some categories such as QUALITY (data parameters measured) or DATA TYPE (nature or genre of the primary data). Here, it should be discussed in the research community how they could be further considered in search, e.g., re-naming or merging of categories. Since the backgrounds of the annotators were quite diverse and no training took place, we did not expect completeness and perfect agreement. We wanted to get a real and unbiased first picture of biodiversity scholars' comprehension when looking for scientific data. In biology and biodiversity research, scholars use a specialized language with diverse content and imprecise and inconsistent naming [31, 32]. Hence, labeling and extracting biological entities remain a challenge.

Concerning the shortened methodology for the evaluation, our assumptions have been confirmed. All nine domain experts completed the classification task. It saved a lot time that only a few people did the identification of artifacts to be labeled and that domain experts assigned categories, only. On average, domain experts spent between two and three hours for labeling 169 questions. We conclude that our shortened annotation approach is fine for opening up new domains and getting insights in what scholars are interested in. If the aim is to achieve higher agreement per annotation, we recommend training sessions and trial rounds. However, such methods may result in biased feedback.

For further reuse of the annotated question corpus, our analysis script also produces an XML file with all questions and annotations above a certain agreement threshold that can be set as a parameter. By default, all annotations per question with an agreement above 0.6 will be returned.

## B—Metadata standards in the life sciences

In this section, we describe a selection of existing metadata standards and investigate whether their elements reflect the identified information categories.

## Methodology

Metadata can be considered as additional information about primary data such as experiments, tabular data, images, sound and acoustic files to make scientific data understandable and machine-readable. In order to enable machine processing, metadata in the Life Sciences are mostly stored in structured formats such as XML or JSON. A *metadata schema* [56] formally describes the syntax and structure of metadata, e.g., which elements and attributes can be used and which elements are mandatory and/or repeatable. If a schema has become established in a research community, it is usually adopted as a *metadata standard*. Another opportunity to become a metadata standard is the formal adoption by a standards' organization such as the International Organization for Standardization, https://www.iso.org. In this work, we use the term 'metadata standard' in a liberal way and include both schemata being adapted by the community and by an organization. We use the term 'metadata field' when discussing individual elements of a concrete metadata file. The term 'metadata structure' comprises all used fields in a metadata file.

As the outcome of the question analysis reveals (Section "A—Information Needs in the Biodiversity Domain"), biodiversity research is a broad field spanning a variety of disciplines. Therefore, in the following analysis, we selected metadata standards from two sources (re3data [57] and RDA Metadata Standards Catalog (version 2) [58]) with a broader perspective. In particular, we wanted to focus on metadata standards being used in current data portals and archives. In re3data, we filtered for "Life Sciences" (https://www.re3data.org/search?query= &subjects%5B%5D=2%20Life%20Sciences). Under the facet 'Metadata standards', a list of 24 standards appeared that we used for our further analysis. From RDA Metadata Standards Catalog, we selected all top-level standards (https://rdamsc.bath.ac.uk/scheme-index) labeled with "Science" (https://rdamsc.bath.ac.uk/subject/Science) resulting in a list of 30 standards. We merged both lists and cleaned the final outcome according to the following criteria: The categories *Other* and *Repository-Developed Metadata Schema* have been omitted. The *MIBBI* standard is outdated and has been integrated into *ISA-Tab*, so we left it out, too. The same applies to the *Observ-OM* and *CIM* standard. The information on the website is deprecated and not fully available (e.g., dead links). We also omitted the "Protocol Data Element Definitions" (data elements that are required for data archival for clinical trials), all astronomy and astrophysics related standards and standards for social and behavioral studies, as these fields are out of our scope. Table 2 presents the final list of 21 metadata standards used for the following analysis.

They are ranked by the number of data repositories supporting them (obtained from re3data). We analyzed whether the standard supports semantic web formats, e.g., RDF or OWL. According to the FAIR principles [11], community standards, semantic formats, and ontologies ensure interoperability and data reuse. The last column provides some examples of data repositories supporting the standard. Further, supplementary material on the selection process can be found in our GitHub repository (https://github.com/fusion-jena/ QuestionsMetadataBiodiv/tree/master/metadataStandards).

The standard supported by most repositories is *Dublin Core*, a general metadata standard based on 15 fields, such as contributor, coverage, creator, date, description, format, and identifier. In addition, data repositories utilize further domain-specific standards with richer vocabulary and structure such as *ISO19115* for geospatial data or *EML* for ecological data. The *RDF Data Cube Vocabulary* is not used by any of the data centers. We suppose, the abbreviation *RDF DC* might lead to a misunderstanding (*Dublin Core* instead of *RDF Data Cube*). All standards provide elements that can be described along the questions: Who? What? Where? When? Why? and How?. In particular, contact person, collection or publication date and

**Table 2. Metadata standards in the (life) sciences obtained from re3data [57] and RDA metadata standards catalog [58].** The number in brackets denotes the number of repositories supporting the standard (provided in re3data).

| Standard Name | URL | Domain | Semantic Format | Examples |
|---|---|---|---|---|
| *Dublin Core* (205) | http://dublincore.org/ | general research data | Yes (RDF) | Pangaea, Dryad, GBIF, Zenodo, Figshare |
| *DataCite* (111) | https://schema.datacite.org/meta/kernel-4.1/ | general research data | No | Pangaea, Zenodo, Figshare, Radar |
| *ISO19115* (47) | ftp://ftp.ncddc.noaa.gov/pub/Metadata/Online_ISO_Training/Intro_to_ISO/workbooks/MD_Metadata.pdf | geospatial data | No | Pangaea, NSF Arctic Data Center, coastMap |
| *FDGC/ CSDGM* (42) | https://www.fgdc.gov/metadata/csdgm-standard | geographic information | No | Dataverse, NSF Arctic Data Center |
| *Darwin Core* (28) | https://dwc.tdwg.org/ | biodiversity data | Yes (RDF) | GFBio, GBIF, VerNET, Atlas of Living Australia, WORMS |
| *EML* (26) | https://knb.ecoinformatics.org/external//emlparser/docs/eml-2.1.1/index.html | ecological data | No | GBIF, GFBio, SNSB, Senckenberg, WORMS, NSF Arctic Data Center |
| *RDF Data Cube* (20) | https://www.w3.org/TR/vocab-data-cube/ | statistical data | Yes | Dryad (only RDF with DublinCore) |
| *ISA − Tab* (13) | https://isa-specs.readthedocs.io/en/latest/index.html | biological experiments | Yes | Data Inra, GigaDB |
| *ABCD* (11) | https://github.com/tdwg/abcd | biological collection data | Yes (ABCD 3.0 in RDF) | GBIF, BioCase Network |
| *OAI − ORE* (11) | https://www.openarchives.org/ore/ | general research data | Yes (RDF) | Environmental Data Initiative Repository, Dryad |
| *DCAT*(9) | https://www.w3.org/TR/vocab-dcat | data catalogs, data sets | Yes | Data.gov.au, European Data Portal |
| *DIF*(9) | https://gcmd.nasa.gov/DocumentBuilder/defaultDif10/guide/index.html | geospatial metadata | No | Pangaea, Australian Antarctic Data Center, Marine Environmental Data Section |
| *CF*(7) | http://cfconventions.org/ | climate and forecast | No | WORMS, NSF Arctic Data Center, coastMap |
| *SDMX* (4) | https://sdmx.org/ | statistical data | No | UN Data Catalog |
| *CSMD − CCLRC* (1) | http://icatproject-contrib.github.io/CSMD/ | scientific activities, e.g., experiments, observations, simulations | Yes (OWL) | Monash University Research Repository |
| *Genome Metadata* (1) | https://docs.patricbrc.org/user_guides/organisms_taxon/genome_metadata.html | genome data | No | PATRIC |
| *OME − XML* | https://docs.openmicroscopy.org/ome-model/6.2.0/ | biological imaging | Yes (OWL) | CELL Image Library, JCB Dataviewer |
| *PDBx/mmCIF* | http://mmcif.wwpdb.org/dictionaries/mmcif_pdbx_v40.dic/Index/index.html | 3D structures of proteins, nucleic acids, and complex assemblies | No | wwPDB (Worldwide Protein Data Bank) |
| *UKEOF* | http://www.ukeof.org.uk/schema/ | environmental monitoring | No | UKEOF catalogue |
| *CEDAR* | https://more.metadatacenter.org/tools-training/outreach/cedar-template-model | general but focus on scientific experiments | Yes (RDF) | US National Cancer Institute's caDSR |
| *Observation and Measurement (OM)* | https://www.ogc.org/standards/om | observational data | Yes (OWL-Lite) | INSPIRE (Infrastructure for Spatial Information in Europe) |

location are considered with one or several metadata fields in all standards. In order to describe the main scope of the primary data, all standards offer numerous metadata fields but differ in their granularity. While simple ones such as *Dublin Core* only offer fields such as title, description, format, and type, standards with more elements such as *EML* or *ABCD* even offer fields for scientific names, methods and data attributes measured. *EML* even allows scholars to define the purpose of the study making it the only standard that supports the *Why* question. Data reuse and citation also play an important role. As it is demanded by the Joint Declaration of Data Citation Principles [59] and practical guidelines for data repositories [60], all standards

provide several elements for digital identifiers, license information and citation. In addition, some standards provide elements for data quality checks. For instance, *ISO19115* offers a container for data quality including lineage information and *EML* supports quality checks with the `qualityControl` element. Increasingly, semantic formats are supported. While some standards already provide or are currently developing fully semantic standards (*ABCD*, *Darwin Core*, *OAI-ORE*, *OME-XML* and *CEDAR*), others provide elements to link taxonomies (*Genome Metadata*, *PDBx/mmCIF*) or started integrating controlled vocabularies (*ISA-Tab*, *UKEOF*, *CEDAR*).

## Results

In our second analysis, we compared the information categories with elements of the metadata standards to figure out, if search interests can be explicitly described with metadata elements. Our results are presented in Table 3. The detailed matching between categories and metadata fields is available as csv file in our *GitHub* repository. For the sake of completeness, we explored all 13 categories from the previous analysis but marked the ones with an asterisk that had a fair agreement ($< 0.4$). The categories are sorted by frequency from left to right. The red color denotes that no element is available in the standard to express the category, orange indicates that only a general field could be used to describe the category and a yellow cell implies that one or more elements are available in the standard for this search interest.

**Table 3. Comparison of metadata standards and information categories.** The categories are sorted by the frequency of their occurrence determined in the previous question analysis, the asterisk denotes the categories with an agreement less than 0.4.

| | Environment | Quality* | Material | Organism | Process | Location | Data Type* | Method* | Anatomy* | Human Intervention* | Event* | Time | Person |
|---|---|---|---|---|---|---|---|---|---|---|---|---|---|
| *DublinCore* | Orange | Orange | Orange | Orange | Orange | Yellow | Yellow | Orange | Orange | Orange | Orange | Yellow | Yellow |
| *DataCite* | Orange | Orange | Orange | Yellow | Orange | Yellow | Yellow | Orange | Orange | Orange | Orange | Yellow | Yellow |
| *ISO19115* | Yellow | Orange | Yellow | Yellow | Orange | Yellow | Yellow | Orange | Orange | Orange | Orange | Yellow | Yellow |
| *FDGC/CSDGM* | Orange | Yellow | Yellow | Yellow | Yellow | Yellow | Yellow | Yellow | Orange | Orange | Orange | Yellow | Yellow |
| *EML* | Yellow | Yellow | Orange | Yellow | Orange | Yellow | Yellow | Yellow | Orange | Yellow | Orange | Yellow | Yellow |
| *DarwinCore* | Yellow | Yellow | Red | Yellow | Red | Yellow | Yellow | Yellow | Red | Red | Red | Yellow | Yellow |
| *RDFDataCube* | Orange | Yellow | Yellow | Yellow | Yellow | Yellow | Yellow | Yellow | Orange | Yellow | Yellow | Yellow | Yellow |
| *ISA – Tab* | Yellow | Yellow | Yellow | Yellow | Yellow | Yellow | Yellow | Yellow | Orange | Red | Red | Yellow | Yellow |
| *DIF* | Orange | Yellow | Yellow | Yellow | Yellow | Yellow | Yellow | Yellow | Orange | Orange | Orange | Yellow | Yellow |
| *CF* | Yellow | Yellow | Red | Red | Yellow | Yellow | Red | Red | Red | Red | Red | Red | Red |
| *ABCD* | Orange | Yellow | Red | Yellow | Red | Yellow | Yellow | Yellow | Orange | Red | Yellow | Yellow | Yellow |
| *DCAT* | Orange | Orange | Orange | Orange | Orange | Yellow | Yellow | Orange | Orange | Orange | Orange | Orange | Yellow |
| *OAI – ORE* | Orange | Yellow | Yellow | Yellow | Yellow | Yellow | Yellow | Orange | Orange | Orange | Orange | Yellow | Yellow |
| *SDMX* | Orange | Yellow | Yellow | Yellow | Yellow | Yellow | Yellow | Yellow | Orange | Yellow | Yellow | Yellow | Yellow |
| *Genome Metadata* | Yellow | Yellow | Red | Yellow | Red | Yellow | Red | Yellow | Yellow | Red | Red | Yellow | Yellow |
| *OME-XML* | Orange | Orange | Yellow | Yellow | Yellow | Yellow | Orange | Orange | Orange | Orange | Orange | Yellow | Yellow |
| *PDB/mmCIF* | Yellow | Yellow | Yellow | Yellow | Red | Red | Yellow | Yellow | Orange | Red | Yellow | Red | Yellow |
| *UKEOF* | Yellow | Orange | Yellow | Yellow | Orange | Yellow | Yellow | Yellow | Orange | Orange | Orange | Yellow | Yellow |
| *CSMD-CCLRC* | Yellow | Orange | Yellow | Yellow | Orange | Yellow | Yellow | Yellow | Orange | Orange | Orange | Yellow | Yellow |
| *CEDAR* | Orange | Orange | Orange | Orange | Orange | Orange | Orange | Orange | Orange | Orange | Orange | Orange | Orange |
| *OM* | Yellow | Yellow | Yellow | Yellow | Orange | Yellow | Yellow | Yellow | Orange | Yellow | Yellow | Yellow | Orange |

Table key

Red: Not provided, Orange: Unspecific (general element), Yellow: Available (one or more elements)

There is no schema that covers all categories. Since the interests are obtained from scholars with various and heterogeneous research backgrounds, this was to be expected. Apart from HUMAN INTERVENTION, all categories are covered by different metadata standards. In particular, *ISA-Tab* covers most of the search interests of biodiversity researchers explicitly. The description of QUALITY information (e.g., data parameters measured) are covered by a variety of standards, e.g., *ABCD*, *Genome Metadata*, *SDMX*. The same applies for information on the DATA TYPE and geographic information (LOCATION). For the latter, the granularity differs between the individual standards. While standards such as *ISO19115*, *ABCD*, *Genome Metadata*, *Data Cite* offering several metadata fields for the description of geographic information, other standards only provide one metadata element, e.g., *CSMD*, *EML*, *Dublin Core*. Metadata elements for environmental information are provided by *EML* (`studyAreaDe-scription`), *UKEOF* (`environmentalDomain`) or *CSDM* (`Facility`). Observed or involved organisms can be described with *Darwin Core*, *ABCD* or *EML* but also with standards focusing on molecular data (*ISA-Tab*, *Genome Metadata* and *PDB/mmCIF*).

However, important search preferences such as materials (including chemicals) are only explicitly supported by *ISA-Tab*, *PDB/mmCIF*, *CSMD* and *OM*. While numerous standards already support the description of research methods and instruments (METHOD), only *ISA-Tab* offers a metadata field (`Protocol REF`) to describe biological or chemical processes (PROCESS) a specimen or an experiment is involved. Even if some categories are not frequently mentioned (Subsection "Results and Metrics"), such as HUMAN INTERVENTION and EVENT, they should not be neglected. Extreme weather events or incidents, e.g., earthquakes or oils spills, might be of research interest and therefore, metadata elements should be provided to define and describe them.

Widely used general standards such as *Dublin Core* or *DataCite* offer at least a general field (`dc:subject`, `subject`) that could be used to describe the identified search categories. In *Dublin Core*, at least one metadata field each is provided to describe geographic information, e.g., where the data have been collected (LOCATION), the type of the data (DATA TYPE), the creator and contributor (PERSON & ORGANIZATION) and when it was collected or published (TIME). However, one field is often not enough to distinguish if the provided field is for instance a collection date or publication date, or if the creator of the dataset is also the person that collected the data. In contrast, *DataCite* provides individual fields for publication year and the `date` field can be used with `dateType="Collected"` to specify a collection date. The metadata field `contributor` can also be extended with a type to indicate whether the contact details belong to the data collector or the project leader. Bounding box elements are also provided to enter geographic coordinates (LOCATION). A complete generic approach is *CEDAR* offering generic elements to set up metadata templates in various formats (e.g., JSON, RDF) and options for integrating controlled vocabularies or *schema.org* entities.

**Summary.** Generic metadata standards such as *Dublin Core* or *DataCite* do not describe primary data sufficiently for dataset search. They cover main scholarly search interests only if general fields are used and proper keywords (in best-case using controlled vocabularies) are contained. A variety of discipline-specific standards are available that go beyond data citation aspects and cover most of the important topics for search, e.g., *ISA − Tab* (biomedical experiments), *Darwin Core*, *ABCD* (species), *EML* (ecology), *ISO19115* (geospatial data), *CSMD*, *PDB/mmCIF* and *ISA − Tab* (materials). However, PROCESS is currently only explicitly covered by *ISA-Tab*. Here, the respective research communities should consider to add elements to the standards that allow an explicit description or at least to provide general fields that allow an extension.

The question that still remains to be answered is what metadata standards are actually used by data repositories.

## C—Metadata usage in selected data repositories

In the following analysis, we examine how well existing metadata cover scholarly information needs in biodiversity research. Therefore, we analyzed metadata from selected data repositories along the following criteria:

- Statistics: At first, we inspected what metadata standards are offered in the repositories' metadata API. For the further processing, we only selected standards that are relevant for biodiversity research.

- Timelines: Timelines indicate how many datasets per year are published in a data repository and when particular standards have been introduced.

- Metadata Field Usage: In order to get an overview of which metadata elements are used we determined how many fields of a standard are filled.

- Category-Field-Match: Focusing on the standard that best covers the identified categories, we explored what fields reflect information needs for biodiversity research and how often they are filled.

- Content Analysis: Counting alone does not provide further insights on the actual content in the metadata. Therefore, we analyzed free text metadata fields such as title, description and further general fields, and we inspected whether they contain useful information for search. Keywords in general fields were extracted and sorted by frequency for all parsed files and from all data repositories. A deeper content analysis of the descriptive metadata fields was limited to 10,000 files per data repository due to performance limitations of the used text mining tools.

We first introduce the methodology comprising data collection, data processing and the used strategy and tools for the content analysis. Afterwards, we present the results along the introduced criteria.

### Methodology

Scholarly publishers increasingly demand scientific data to be submitted along with publications. Since publishers usually do not host the data on their own, they ask scholars to publish data with a repository for their research domain. According to *Nature's* list of recommended data repositories [16], we selected five archives for our further analysis: three generalist ones (*Dryad*, *Figshare* and *Zenodo*) and two domain-specific ones (*PANGAEA*—environmental data, *GBIF*—taxonomic data). In the biodiversity projects we are involved in, scholars also often mention these repositories as the ones they mainly use.

**Data collection.** We parsed all available metadata from *Figshare*, *Dryad*, *GBIF*, *PANGAEA* and *Zenodo* in May 2019 via their respective *OAI-PMH* (https://www.openarchives.org/pmh/) interfaces. The Open Archives Initiative Protocol for Metadata Harvesting (OAI-PMH) is a protocol primarily developed for providing and consuming metadata. *GBIF* only offers the metadata of their datasets in the *OAI-PMH* interface. The individual occurrence records, which belong to a dataset and which are provided in metadata structures based on the *Darwin Core* metadata standard [61], are not present in the *OAI-PMH* interface. Hence, we only analyzed the metadata of the datasets.

Our script parses the metadata structure of all public records per metadata standard for each of the selected data repositories (Table 4). Apart from the metadata standards introduced in the previous section, a few more standards appear in this list. *OAI-DC* is an abbreviation for *Dublin Core*, a mandatory standard in OAI-PMH interfaces. *QCD* means *qualified Dublin*

**Table 4. Metadata schemes and formats offered by selected data repositories in their OAI-PMH interfaces.**

| Dryad | GBIF | PANGAEA | Zenodo | Figshare |
|---|---|---|---|---|
| METS | EML | DATACITE3 | DATACITE | CERIF |
| OAI-DC | OAI-DC | DIF | DATACITE3 | METS |
| OAI-ORE | | ISO19139 | DATACITE4 | OAI-DATACITE |
| RDF | | ISO19139.IODP | MARCXML | OAI-DC |
| | | OAI-DC | MARC21 | QDC |
| | | PAN-MD | OAI-DATACITE | RDF |
| | | | OAI-DATACITE3 | |
| | | | OAI-DC | |

*Core* and denotes an extended *Dublin Core* extending or refining the 15 core elements. We also considered *Pan-MD*, a metadata schema developed by *PANGAEA*. It extends *Dublin Core* with more fine-grained geographic information such as bounding boxes or adds information on data collection. The latter can range from projects, parameters, methods, and sensors to taxonomy or habitats.

*MARC21*, *MARCXML* and *METS* are metadata standards that are mainly used for bibliographic data in digital libraries. Hence, we left them out of our further explorations. We also did not consider the *Common European Research Information Format (CERIF)* and *OAI-ORE* as they are not focused on describing primary data but research entities and their relationships and grouping web resources, respectively. However, we decided to permit all available repository-developed schemata for the Life Sciences, such as *Pan-MD*, in order to get an impression how repositories extend metadata descriptions. *RDF* is not a standard but a semantic format. We decided not to omit it as it indicates the usage of semantic formats at data repositories. The metadata fields used in RDF format are mainly based on *Dublin Core*.

**Data processing.** Per metadata file we inspected which elements of the metadata standards are used, and we saved their presence (1) or non-presence (0). The result is a csv file per metadata standard containing dataset IDs and metadata elements used. All generated files are stored in separate folders per repository and metadata standard. Each request to a repository returns an XML body that includes several metadata files as records. Each record is separated in two sections, a header and a metadata section. The header section comprises general information such as a unique identifier of the item and a date stamp. The metadata section contains the metadata structure, e.g., the name of the contributors, abstract and publication year. Unused metadata fields are not included in the response. We saved a boolean value encoding whether a metadata field was used or not. The source code and documentation on how to use it is available in our GitHub repository.

For our further consideration, we wanted to obtain a publication date of each downloaded dataset to inspect how many datasets have been published over the years in which standard per data repository. Unfortunately, a publication date is not provided in all metadata schemata. Therefore, we looked up each date related field in the schema and used the one that is (based on the description) the closest to a publication date. Table 5 depicts all date stamps utilized and their descriptions. If the respective date stamp was not found in a dataset or was empty, we left the dataset out in the following analysis.

**Data collection for content analysis.** General, descriptive metadata fields such as 'title', 'description' or 'abstract', and 'subject' might contain relevant data that are interesting for information seekers. Using conventional retrieval techniques, this data is only accessible in a full text search and if the entered query terms exactly match a term in the dataset. These

**Table 5. The date stamps used for each metadata standard and their descriptions obtained from the standard's website.**

| Format | Element | URL | Description |
|---|---|---|---|
| *OAI-DC/METS/QDC RDF(Dryad)* | dc:date | http://www.dublincore.org/specifications/dublin-core/dces/ | "A point or period of time associated with an event in the lifecycle of the resource." |
| *EML* | pubDate | https://knb.ecoinformatics.org/external//emlparser/docs/eml-2.1.1/eml-resource.html#pubDate | "The 'pubDate' field represents the date that the resource was published." |
| *DATACITE3/4 OAI-DATACITE/3* | publicationYear | https://support.datacite.org/docs/schema-40 | "The year when the data was or will be made publicly available." |
| *DIF* | DIF_Creation-_Date | https://gcmd.gsfc.nasa.gov/DocumentBuilder/defaultDif10/guide/metadata_dates.html | "refers to the date the data was created" |
| *ISO19139/ISO19139.iodp* | gco:DateTime | https://geo-ide.noaa.gov/wiki/index.php?title=ISO_Dates | (CI-DataTypeCode=publication), publication Date |
| *PAN-MD* | md:dateTime | http://ws.pangaea.de/schemas/pangaea/MetaData.xsd | publication date (contact to data repository) |
| *RDF(Figshare)* | vivo:datePublished | https://codemeta.github.io/terms/ | "Date of first broadcast/publication" |

metadata fields potentially contain the most relevant search terms and are also candidates to be enriched with text mining tools. Hence, we aim to explore what information is currently available in general, descriptive metadata fields. We downloaded descriptive metadata fields, namely, `dc:title`, `dc:description` and `dc:subject` in *OAI-DC* format from all repositories in October and November 2019. Parallel to the download, we collected the key-words used in the subject field and counted their presence in a separate csv file.

In order to further inspect the content with Natural Language Processing (NLP) [24] tools, we selected a subset of representative datasets. We limited the amount to 10,000 datasets per repository as the processing of textual resources is time-consuming and resource-intensive. A variety of applications have been developed to determine Named Entities (NE) such as geo-graphic locations, persons and dates. Thessen et. al [31] explored the suitability of existing NLP applications for biodiversity research. Their outcome reveals that current text mining sys-tems, which were mainly developed for the biomedical domain, are able to discover biological entities such as species, genes, proteins and enzymes. We concentrated on the extraction of entity types that (a) correspond to the identified search interests (Section "A—Information Needs in the Biodiversity Domain") and for which (b) text mining pipelines are available. We used the text mining framework GATE [62] and its ANNIE pipeline [63] as well as the Orga-nismTagger [64] to extract geographic locations, persons, organizations and organisms. In our previous research, we developed the BiodivTagger [65], a text mining pipeline extracting envi-ronmental terms, materials, processes and data parameters.

## Results

In this subsection, we present the results of the harvested and processed metadata. We ana-lyzed the data along the introduced criteria, beginning with statistics and timelines, a study on the use of metadata fields and a deeper content analysis of metadata fields.

**Statistics.** The overall statistics of the analyzed data repositories are presented in Table 6. Most repositories support general standards for metadata creation; only *PANGAEA* and *GBIF* utilize metadata based on discipline-specific metadata standards. *Dryad* and *Figshare* already provide metadata in semantic formats such as RDF. In addition, *Figshare* offers *Qualified Dub-lin Core (QDC)*, an extended *Dublin Core* that allows the description of relations to other data sources. We checked the fields for a publication date in a valid format. We could not use all harvested datasets as for some metadata files publication dates were not available. *Dryad* had a

**Table 6. Total number of datasets parsed per data repository and metadata standards and schemata.** The numbers in brackets denote the number of datasets used for the analysis. All datasets were harvested and parsed in May 2019.

| Metadata Standard | Dryad | PANGAEA | GBIF | Zenodo | Figshare |
|---|---|---|---|---|---|
| *OAI-DC* | 186951 (142329) | 383899 (383899) | 44718 (42444) | 255000 (255000) | 3128798 (3128798) |
| *QDC* | | | | | 1718059 (1718059) |
| *RDF* | 186955 (142989) | | | | 3157347 (3157347) |
| *DATACITE* | | | | 1268155 (1268155) | |
| *DATACITE3* | | 383906 (383906) | | 1268232 (1268232) | |
| *OAI-DATACITE* | | | | 1266522 (1266522) | 3134958 (3134958) |
| *OAI-DATACITE3* | | | | 1268679 (1268679) | |
| *DATACITE4* | | | | 1268262 (1268262) | |
| *EML* | | | 44718 (42444) | | |
| *DIF* | | 383899 (383899) | | | |
| *ISO19139* | | 383899 (383899) | | | |
| *ISO19139.iodp* | | 383899 (383899) | | | |
| *PAN-MD* | | 383899 (383899) | | | |

large number of datasets with a status "Item is not available", which we left out, too. The number in brackets denotes the amount of datasets we used for the following considerations.

**Timelines.** Based on the given publication dates, we computed timelines (Fig 5) for the introduction of the various standards over time per data repository. The code and all charts are available as supplementary material in our *GitHub* repository. As *Dryad* provides several `dc:date` elements in the metadata, we used the first available date entry as publication date for the timeline chart.

Per repository, the timelines for the different metadata formats are almost identical. Obviously, when introducing a new metadata format, publication dates were adopted from existing metadata formats. Only *Figshare* uses new date stamps when a new metadata format is provided. For instance, *Figshare*'s timeline shows that QDC and RDF were launched in 2015. The result for RDF was too large to process it together with the other metadata formats. Hence, we produced the timeline for RDF separately. The timelines across all repositories reveal a steadily increasing number of datasets being published at *GBIF*, *Dryad*, *Zenodo* and *Figshare*. For *PANGAEA*, the timeline points to a constant number of published datasets of around 10,000 datasets a year apart from an initial release phase between 2003 and 2007.

**Metadata field usage.** Fig 6 presents how many metadata elements of the best matching standard were filled. The individual results per data archive are available in our repository as supplementary material (https://github.com/fusion-jena/QuestionsMetadataBiodiv/tree/master/data_repositories/charts). *Dryad* used 9 out of 15 available metadata fields from *OAI-DC* very often (> 80%) including important fields such as `dc:title`, `dc:description` and `dc:subject`. `dc:publisher` and `dc:contributor` were provided in less that 20%. For *GBIF*, the *EML* standard does not provide a fixed number of core elements. Hence, we analyzed the 129 available fields. Most of them (89 elements) were not filled, e.g., fields describing taxonomic information. Data about author, title and description were provided in more than 80%. The general field `eml:keyword` was used in around 20%. Out of 124 used fields in *PANGAEA*'s Pan-MD format, 43 fields were filled in more than 80% of the harvested metadata files including information on the author, project name, coordinates, data parameters and used devices. Fields that were less filled are supplementary fields, for instance for citation, e.g., volume, pages. For *Zenodo*, all required fields in *DataCite* (`identifier, creator, title, publisher, publication year`) were always filled. In

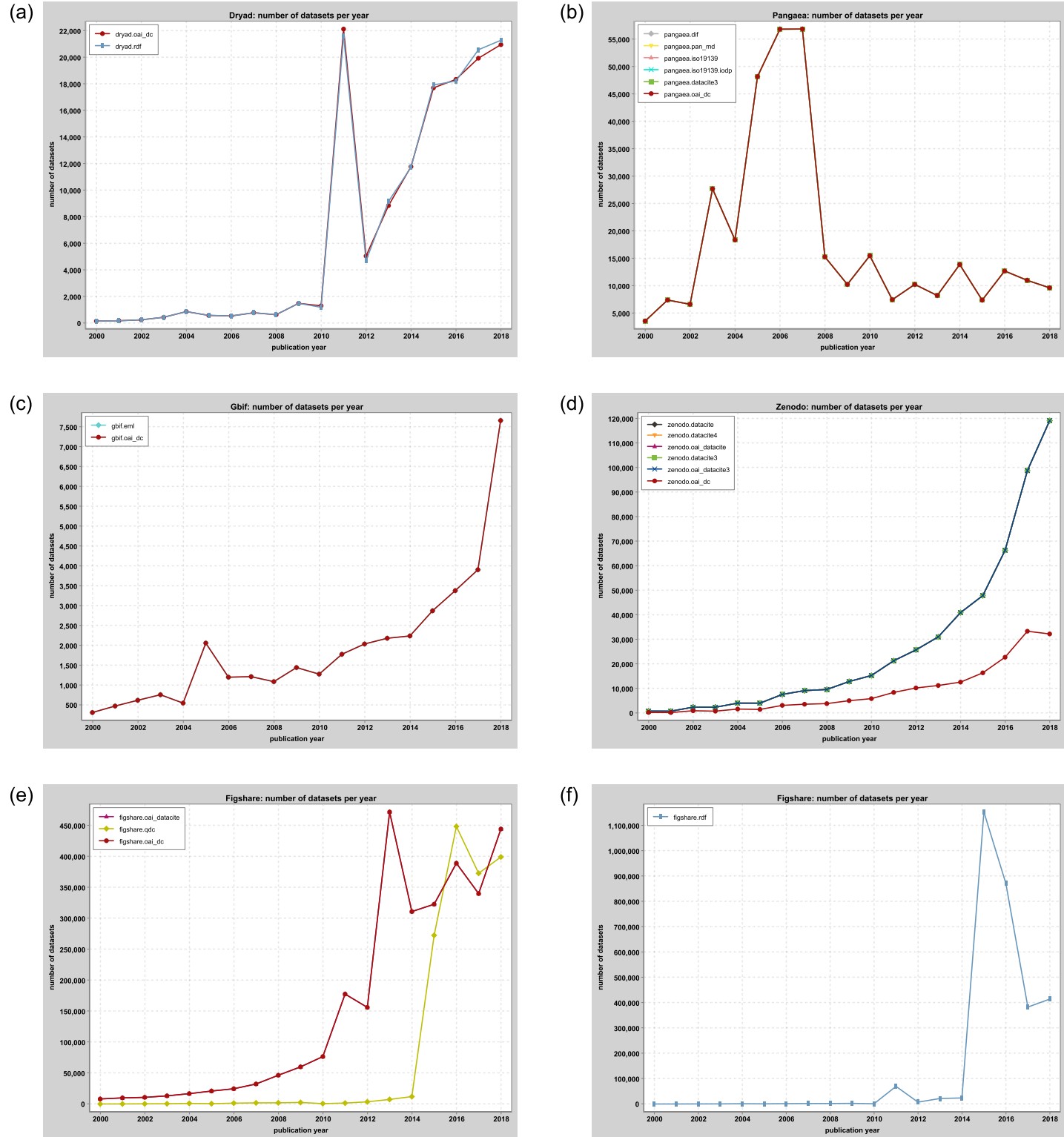

**Fig 5. Timelines for all repositories presenting the number of datasets per metadata standard and schema offered.** For several repositories, the timelines for the different metadata standards and schemata are almost identical and overlap. Obviously, when introducing a new metadata standard or schema, publication dates were adopted from existing metadata structures. *Figshare's* timeline for RDF was computed separately as the data are too large to process it together with the other metadata files.

(a)
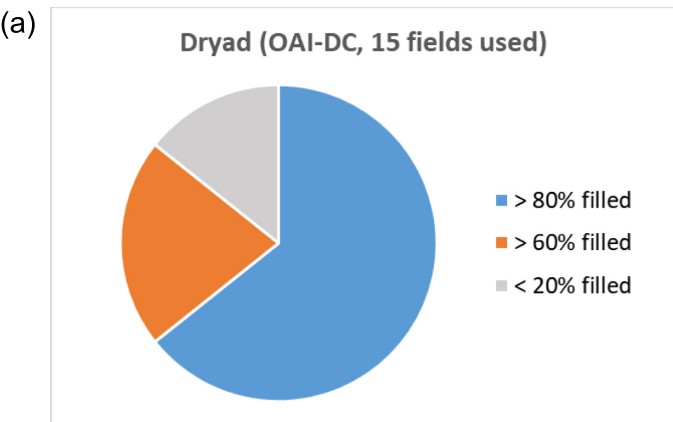

(b)
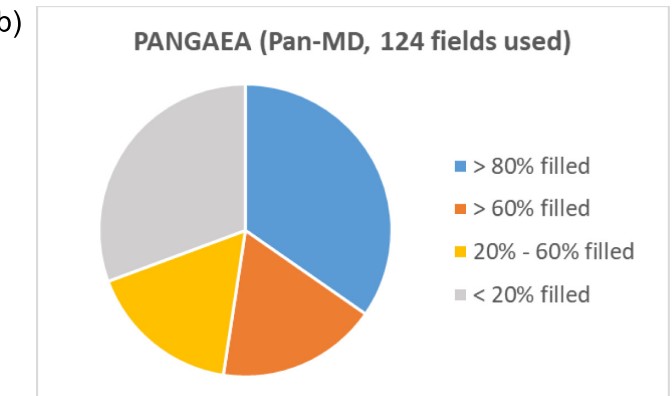

(c)
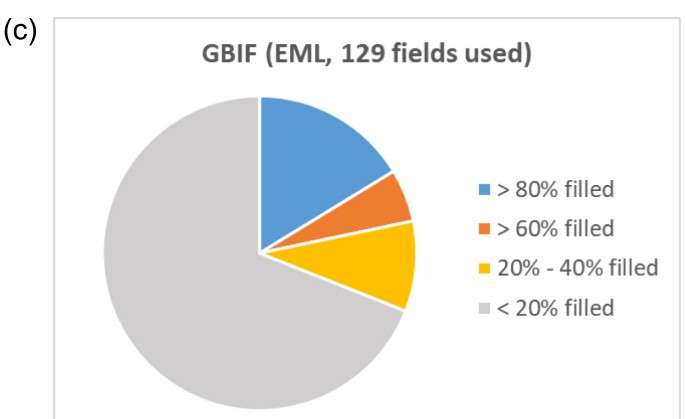

(d)
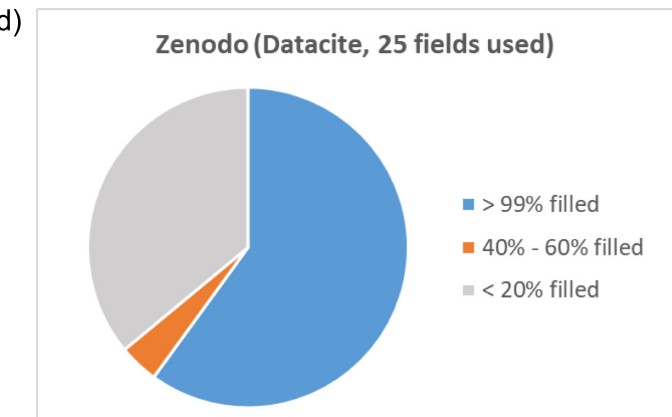

(e)
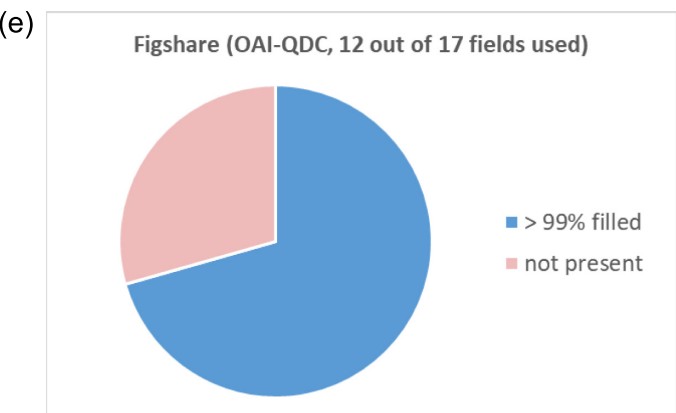

**Fig 6. Metadata field usage in all data repositories evaluated.** The graphics display the percentage of metadata fields used per data repository and its best matching standard with respect to the information categories.

addition, title, rights and descriptions as well as resource type were also provided in more than 99% of the analyzed metadata files. However, in only 45% of the metadata files, keywords (`subject`) were present. *Figshare* used only 12 out of 17 available fields of *QDC*, but these fields were always filled. 5 out of 17 fields of *QDC* were not used at all.

**Table 7. Comparison of data repositories and their best matching standard with the information categories.** The categories are sorted by the frequency of their occurrence determined in the question analysis. The asterisk denotes the categories with an agreement less than 0.4.

| | Environment | Quality* | Material | Organism | Process | Location | Data Type* | Method* | Anatomy* | Human Intervention* | Event* | Time | Person |
|---|---|---|---|---|---|---|---|---|---|---|---|---|---|
| *GBIF* (*EML*) | (3%) | | | (11%) | | (35%) | (8%) | (18%) | | | | (publication date—100%, collection data—10%) | (>90%) |
| *Dryad* (*OAI – DC*) | | | | | | (60%) | | | | | | (publication date) | (80%) |
| *PANGAEA* (*Pan – MD*) | | (>90%) | | | | (100%) | (100%) | (Devices used—90%, research methods—65%) | | | | (publication Date—100%, collection date—80%) | (100%) |
| *Zenodo* (*OAI – Datacite*) | | | | | | | (100%) | | | | | (publication Date) | (100%) |
| *Figshare* (*QDC*) | | | | | | | (100%) | | | | | (publication Date) | (100%) |

Table key

Orange: Unspecific (generic element), Yellow: Available (one or more elements)

Amount in brackets denotes the percentage the element is filled.

**Category—Field—Match.** Per data repository and metadata standard, we computed charts that visualize which field was filled at what percentage rate and if they correspond to the categories introduced in Section "A—Information Needs in the Biodiversity Domain". Table 7 presents a summary of all data repositories and their best matching standard. The individual results per repository and the concrete field-to-category mapping are available in our repository.

Temporal expressions (TIME) and information about author and/or creator (PERSON) were mostly provided in all repositories. Apart from *PANGAEA*, repositories mainly provided the publication date and only partially added information about when the data was collected. Information about data type and formats was also contained in all metadata files apart from *GBIF*. The identified search categories were partially covered by two repositories. *GBIF* with *EML* reflects most of the categories, but fields that correspond to ENVIRONMENT, ORGANISM, DATA TYPE and METHOD were rarely filled. Metadata files in *PANGAEA′s* repository-developed standard Pan-MD always contained information on data parameters (QUALITY) and geographic locations. In most cases, research methods and devices used were also given. *Dryad* provided geographic information (LOCATION) in its `dc:coverage` field in at least 60%.

**Content analysis.** Table 8 presents the five most common keywords in the metadata field `dc:subject` for all repositories sorted by their frequencies. The full keyword lists are available in our repository.

For *GBIF* datasets, in 81% an empty `dc:subject` field was returned. *Zenodo's* metadata provided keywords in 52% of the inspected cases. There are inconsistencies in how terms are entered; for example, none of the repositories seem to consider upper and lower cases. For several terms, different spellings resulted in separate entries. Numerous keywords in *PANGAEA*

**Table 8. Five most common keywords and their frequencies in the metadata field `dc:subject`.** The last row denotes the amount of files with an empty `dc:subject` field.

| PANGAEA | GBIF | Dryad | Zenodo | Figshare |
|---|---|---|---|---|
| water (201102) | Occurrence (6510), occurrence (46) | Temperature (16652), temperature (15916) | Taxonomy (459877), taxonomy (105) | Medicine (1057684), medicine (240) |
| DEPTH (198349), Depth (71916) | Specimen (3046), specimen (22) | Integrated Ocean Observing System (16373) | Biodiversity (458336), biodiversity (8593) | Biochemistry (1015906), biochemistry (92) |
| Spectral irradiance (175373) | Observation (2425), observation (24) | IOOS (16373) | Herbarium (270110), herbarium (91) | Biological Sciences not elsewhere classified (983829) |
| DATE/TIME (128917) | Checklist (589), checklist (43) | Oceanographic Sensor Data (15015) | Terrestrial (269900), terrestrial (177) | Chemical Sciences not elsewhere classified (842865) |
| Temperature (118522), temperature (50) | Plantas (368), plantas (42) | continental shelf (15015) | Animalia (205242), animalia (261) | Biotechnology (792223), biotechnology (23978) |
| 0 | 38296 | 15436 | 705730 | 0 |

and *Dryad* reveal that both repositories host marine data. *PANGAEA*'s list mainly contains data parameters measured and used devices. In contrast, *Dryad's* list indicates that terrestrial data are also provided. For instance, the lower ranked terms contain entries such as Insects (1296) (insects (180)) or pollination (471) (Pollination (170)). Geographic information, e.g., California (9817), also occurred in *Dryad's* `dc:subject` field. *Zenodo's* and *Figshare's* keyword lists contain numerous terms related to collection data. We checked the term 'Biodiversity' in both repositories in their search interfaces on their websites. It turned out that the *Meise Botanic Garden* (https://www.plantentuinmeise.be) provided large collection data in *Zenodo*. Hence, each occurrence record counted as a search hit and got the label 'Biodiversity'. We also discovered that *Figshare* harvests *Zenodo* data which also resulted in high numbers for *Figshare* and the keyword 'Biodiversity' (219022).

In a second analysis, we investigated entity types that occur in descriptive metadata fields. As the processing of textual resources with NLP tools is time-consuming and resource-intensive, we selected a subset of datasets. We limited the amount to 10,000 datasets per repository. Table 9 presents the filter strategies. For *PANGAEA* and *GBIF*, we randomly selected 10,000 datasets as they are domain-specific repositories for which all data are potentially relevant for biodiversity research. For *Zenodo* and *Figshare* we used the keyword 'Biodiversity'. Due to the large amount of collection data in *Zenodo* and *Figshare* with the keyword 'Biodiversity', we are aware that this filter strategy may have led to a larger number of collection datasets included in the 10,000 selected files. For *Dryad*, the filter contained a group of relevant keywords as a filter with only one keyword would not have returned at least 10,000 datasets.

Per data repository, we processed the selected 10,000 files with open source taggers of the text mining framework GATE [62]. Named Entities such as PERSON, ORGANIZATION and LOCATION were obtained with the ANNIE pipeline [63], ORGANISMs were extracted with the OrganismTagger [64], and further biological entities (QUALITY, PROCESS, MATERIAL

**Table 9. Filter strategies used per data repository to select 10,000 datasets.** The number in brackets denotes the total number of available datasets (*OAI-DC* standard) at the time of download (October/November 2019).

| PANGAEA | GBIF | Dryad | Zenodo | Figshare |
|---|---|---|---|---|
| 10000 randomly selected (388142) | 10000 randomly selected GBIF (46954) | 10000 randomly with keywords: biodiversity, climate change, ecology, insects, species richness, invasive species, herbivory, pollination, endangered species, ecosystem functioning, birds (149671) | 10000 randomly with keyword: Biodiversity (1457958) | 10000 randomly with keyword: Biodiversity (3592808) |

**Table 10. NLP analysis: Number of datasets with named entities (out of 10,000 processed files in a reduced *OAI-DC* structure) per repository.** Each file contains a subset of the original metadata, namely, `dc:title`, `dc:description`, `dc:subject` and `dc:date`.

| | PANGAEA | GBIF | Dryad | Zenodo | Figshare |
|---|---|---|---|---|---|
| Location | 9088 | 5718 | 3530 | 4644 | **9978** |
| Person | 3789 | **6687** | 3030 | 3201 | 1773 |
| Organization | 3307 | 1762 | 1486 | 1674 | **9542** |
| Environment | 2918 | 1533 | **5166** | 2050 | 15 |
| Quality | **7794** | 1830 | 6548 | 3183 | 29 |
| Process | 1756 | 1055 | **6743** | 1847 | 15 |
| Material | **8593** | 3358 | 3716 | 2859 | 644 |
| Organism | 3251 | 1217 | 3603 | 2891 | **8542** |

and ENVIRONMENT) were obtained from the BiodivTagger [65]. The results are presented in Table 10. ORGANISM annotations are contained in 85% of the analyzed *Figshare* files, in 36% of *Dryad* files, in 32.5% of the selected *PANGAEA* files, in 29% of *Zenodo* files and in around 12% of the selected *GBIF* files. The number of ORGANISM annotations in *GBIF* files is low since datasets mostly describe the overall study and do not contain concrete species names but rather broader taxonomic terms such as 'Family' or 'Order'. Probably, the number of ORGANISM annotations in *Figshare* files is that high due to the large amount of collection data. The numbers for all other biological entity types in *Figshare* are low because, apart from the taxonomic classification, no further relevant keywords are provided in the descriptive fields. While the inspected *PANGAEA* files achieved the highest results for the categories MATERIAL (86%) and QUALITY (78%), *Dryad* provides the largest amount of files containing annotations of the categories ENVIRONMENT (52%) and PROCESS (67%). That shows that the content of descriptive metadata fields actually contains relevant information for search, which can be made accessible through additional semantic enrichment.

Some of the text mining pipelines were originally developed and evaluated with text corpora and not sparse datasets. Hence, annotations might be missing or the results could contain wrong (false positive) annotations. However, the outcome indicates that NLP tools can support the identification of biological entities. That could be an approach for generalist repositories to additionally enrich metadata. All scripts and the final results are available in our repository.

**Summary.** In all repositories, we determined an increasing amount of datasets over the years. In some of them, the growth rate is exponential. This confirms the importance of data repositories and the increasing demand for data publication services. Our results reveal that generalist repositories tend to use only general standards such as *Dublin Core* and *DataCite*. Even if it is common understanding that generalist repositories only support general metadata standards, it would be a great advantage for data discoverability if generalist repositories offer several different metadata standards. This would additionally emphasize the message of generalist repositories to be able to handle any type of scientific data. Discipline-specific repositories, e.g., *GBIF* and *PANGAEA* are more likely to provide domain-related standards such as *EML* or *Pan-MD*. In *GBIF's* case, we are aware that we could not provide a full picture as we did not analyze the occurrence records. Here, only a deeper analysis of the provided fields in the search index would deliver more answers. However, that would require technical staff support as the access to GBIF's search indices is limited.

Overall, the only metadata fields that matched researchers' search interests closely were TIME and PERSON, which were present in almost all inspected repositories. However, important categories such as information on environments, materials or species are only covered if

general fields or longer textual fields such as title and description contain useful keywords. Most repositories indeed enhance metadata with numerous keywords in general fields. They reflect scholarly information needs to some extent. The extracted keyword list emphasizes the heterogeneity of research data in biodiversity research and shows in which different granularity data are described. Keywords vary between broad subject categories and concrete measuring methods. In addition, different spellings amplify the problem of inconsistency.

## D—Discussion

In this study, we explored what hampers dataset retrieval in biodiversity research. The following section discusses our findings, highlights topics that should be further discussed in the biodiversity research community and lists remaining challenges for computer science.

### Scholarly search interests in biodiversity research

We identified five information categories being important for search tasks in biodiversity research, namely, ENVIRONMENT, MATERIAL, ORGANISM, PROCESS and LOCATION. The categories QUALITY and DATA TYPE only got a fair agreement but are mentioned quite often in the search questions.

Comparing our question analysis to the outcome of the content assessment analysis conducted in the *GBIF* community [28] in 2009, the authors' assumption of this study that user interests change over time has been confirmed. Species are still an important category scholars are interested in, however, further important topics for the acquisition and description of ecosystem services are emerging. Our results confirm some of the defined dimensions in EASE [43], a metadata schema developed for search purposes. Common categories are ORGANISM, PROCESS, TIME and LOCATION (Space). Instead of the category QUALITY, data parameters in EASE are collected in sub-elements of the dimension METHOD. For some categories, our definitions go beyond the EASE dimensions. We found that MATERIALs in a broader sense are relevant to biodiversity researchers and not only chemicals. The same applies for environmental terms. They are in general important for dataset retrieval in biodiversity research and not just biomes. In addition to EASE we found that a separate category DATA TYPE should be considered as many researchers asked for specific result types of research methods.

In summary, we conclude that the diversity of scholarly search interests in biodiversity research can indeed be structured and grouped into a limited amount of categories. Considering these interests in dataset descriptions and search interfaces will facilitate dataset retrieval.

### Metadata analysis of selected data repositories

Our study reveals that current metadata in data repositories match scholarly search interests only to some extent. Only if proper keywords are contained in general or descriptive fields, conventional retrieval algorithms are able to find relevant datasets. Poorly described data in combination with keyword-based retrieval techniques that are mainly used in retrieval systems of data repositories [4] result in less relevant hits in search results or a long time spent until suitable data have been retrieved [3, 6, 7]. That might cause users to send explicit data requests to data repositories [5] or to switch to other data sources such as literature and general search engines [8].

Improved filter options such as a faceted search only support data seekers if either relevant, explicit metadata fields are provided or keywords are aligned and grouped [33]. Arbitrary metadata descriptions hamper both. Although metadata are not only created for search purposes, they are a key building block for successful data retrieval. Therefore, scholars need to be

aware that thorough data descriptions are part of a good scientific practice. In order to preserve all kind of scientific data, independently of whether it has been used in publications or not, proper metadata descriptions based on appropriate schemata enhance dataset retrieval and thus, increase data reuse and data citation. Data repositories could offer data curation services to support scholars in describing research data and to encourage them to describe their data thoroughly. We are aware that it would require high efforts to introduce more domain-specific metadata standards at generalist repositories; however, it would improve data descriptions.

## Challenges to be addressed in the biodiversity research community

The importance of metadata for data discovery has already been recognized by the research community. In the FAIR Data Maturity Model [66], a guideline with indicators for the evaluation of the FAIR principles [11], all *Findability* indicators were considered as 'essential'. The second *Findability* indicator (F2-01M) [66] refers to the richness of metadata. It says that an evaluation should consider the amount of metadata and its suitability for data discovery. However, we think, it is unsatisfactory to tell scholars to provide 'rich metadata' without any further guidance, as it is often not clear what information is relevant for data discovery per research domain. Here, we recommend to extend the indicator, e.g., with additional documents per domain, providing more information about the various metadata standards and important search categories per research field. Several studies already proposed guidelines for data repositories [67] and scholars [68, 69] to ensure good data management favoring the usage of domain-specific standards. However, our study reveals that large data repositories only partly follow these recommendations. In particular, general data repositories tend to use only general metadata standards. Although guidelines and proper metadata data standards exist, the question remains why are they so rarely used? Are the recommendations too broad? Are there too many standards or do standards contain too many fields and elements? Do we need improved software tools and methods to facilitate the integration of different metadata in one search index? We think, only further user studies and discussions in the research community may help to find answers on these questions.

First results of such discussions towards more fine-grained recommendations for essential metadata fields in specific research domains are the initiative from *GBIF* [70] and the work of Jurburg et al [71]. Both approaches aim for a better description of sequence data. We think, more in-depth discussions per research field will support data descriptions and metadata creation as relevant search categories are domain dependent. In biodiversity research, a variety of standards are already available. Taxonomic and occurrence data could be comprehensively described with *Darwin Core* or *ABCD*. For environmental data descriptions, *EML* would be a good choice. *EASE* is not a standard yet but could also be considered for future environmental data descriptions. In order to properly describe digital geographic data, *ISO19115* is the recommended standard of the respective research community. Genome data could be thoroughly described using *ISA-Tab*. Moreover, *ISA-Tab* is suitable for the description of a diverse set of environmental and biomedical experiments that employ one or a combination of technologies. For more individual metadata description, but using controlled vocabularies and terminologies, *CEDAR* could be an option.

Our findings reveal that biodiversity data should not only be described from a specific research focus but should contain supplementary information. Apart from observed species, environmental information such as habitat or biome, involved biological, physical and chemical processes, analyzed materials, data parameters measured and geographical information should be considered in metadata creation.

## Challenges to be addressed in computer science

Besides domain-specific metadata standards, the usage of controlled vocabularies is the second important building block for successful dataset retrieval. According to the RDA Data Maturity Model [66], controlled vocabularies ensure interoperability (I2-01M). From computer science perspective, aligned terminologies that ideally provide URIs to concepts in ontologies allow further technical improvements.

Computer science research can contribute to improvements for dataset search by developing methods and software tools that facilitate standard-compliant metadata provision ideally at the time of data collection, thus ensuring metadata standards to be actually used by data providers. Such tools for metadata creation should provide the opportunity to use controlled vocabularies for the data description. In a best case scenario, even the raw data contain aligned terminologies facilitating automatic metadata creation. In the Life Sciences, a variety of controlled terminologies and terminology providers already exist. We provide a list of ontology and semantic service providers in our GitHub repository (https://github.com/fusion-jena/QuestionsMetadataBiodiv/blob/master/biodivTerminologyServices.md).

For legacy data, data structure changes are less feasable and therefore, other computer science techniques are necessary to extract additional information and to improve metadata. The text mining community has already developed various taggers and pipelines to extract organisms [64], chemistry items [62] or genes [72] from textual resources. Recently, a tagger has been developed for biodiversity metadata [65] identifying environmental terms, data parameters, processes and materials. These annotations could support automatic facet or category creation. In addition, some of these taggers already provide a linkage to the Linked Open Data (LOD) (https://www.lod-cloud.net/) Cloud and controlled vocabularies. Our results of the content analysis reveals that text mining can help to overcome the shortcomings of current metadata and support the enrichment of metadata with additional relevant information for data discovery. However, more investigations are needed to determine how many false positive annotations occur and if there is a need to further adapt existing pipelines for metadata enrichment.

Semantic enrichment also allows the integration of *schema.org* (https://schema.org) or *bioschemas.org* (https://bioschemas.org/). Driven by Google and the RDA Data Discovery Paradigm Interest Group, this community initiative aims to provide structured information for HTML web pages such as unique identifiers, persons, locations or time information. That supports external search engines or data providers to crawl the landing pages of search applications provided by the data repositories per dataset.

Finally, with controlled vocabularies and ontological enrichment full semantic search systems are possible. First approaches were mainly developed for publications in bio-medicine and health sciences, e.g., [73–77]. Semantic search systems need controlled vocabularies for proper entity recognition and disambiguation. That enables developers of search systems in data portals to expand a search query on semantically related terms such as synonyms or more specific or broader terms [76, 77] or faceted search [74, 75, 77]. Due to these numerous studies in bio-medicine, large data repositories such as EBI [78] and NCBI [79] already provide semantic techniques in their search systems. As the timelines in Subsection "Results and Metrics" reveal, *Dryad* and *Figshare* also started using semantic formats for metadata but so far do not offer a semantic search. To the best of our knowledge, there are few semantic search approaches focusing on dataset retrieval in biodiversity research [80, 81]. In addition, there are only very few systems that consider semantically related terms or categories in scoring [77]. Hence, as semantic search approaches are still under research in computer science, integration in current data portals has only just started.

## Conclusion

Scholarly search interests are as diverse as data are and can range from specific information needs such as searches for soil samples collected in a certain environment to broader research questions inspecting relationships among species. Our findings reveal that these search interests are not entirely reflected in existing metadata. One problem are general standards that are simple and mainly contain information that support data citation. Important search interests are only be represented if keywords and suitable search terms are provided in general, non-specific fields. Most data repositories utilize these fields to enrich metadata with suitable search terms. However, if search interests are not explicitly given, facet creation, e.g., filtering over species or habitats, is more difficult.

Data findability, one of the four FAIR principles [11], at least partially relies on rich metadata descriptions reflecting scholarly information needs. If the information scholars are interested is not available in metadata, the primary data can not be retrieved, reused and cited. In order to close this gap, we address challenges to overcome the current obstacles for both, the biodiversity research community and computer science.

In our future work, we would like to focus on a machine-supported extraction of relevant search categories in metadata as well as an automatic filling of metadata fields from primary data. That will minimize the metadata creation process and will support scholars and data repositories in producing proper and rich metadata with semantic enrichment.

## Acknowledgments

The authors would like to thank the annotators for their time and valuable comments.

## Author Contributions

**Conceptualization:** Felicitas Löffler, Friederike Klan.

**Formal analysis:** Felicitas Löffler, Friederike Klan.

**Funding acquisition:** Birgitta König-Ries.

**Methodology:** Felicitas Löffler, Friederike Klan.

**Project administration:** Felicitas Löffler.

**Resources:** Felicitas Löffler, Valentin Wesp.

**Software:** Felicitas Löffler, Valentin Wesp, Friederike Klan.

**Supervision:** Birgitta König-Ries.

**Validation:** Felicitas Löffler, Friederike Klan.

**Writing – original draft:** Felicitas Löffler, Valentin Wesp.

**Writing – review & editing:** Felicitas Löffler, Birgitta König-Ries, Friederike Klan.

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
