## [Decision Letter · Decision Letter 0]

30 Sep 2019

PONE-D-19-18843

FAIR data in biodiversity: Are metadata rich and findable?

PLOS ONE

Dear Mrs. Löffler,

Thank you for submitting your manuscript to PLOS ONE. After careful consideration, we feel that it has merit but does not fully meet PLOS ONE’s publication criteria as it currently stands. Therefore, we invite you to submit a revised version of the manuscript that addresses the points raised during the review process.  In particular, both reviewers were skeptical about a central claim in this manuscript relating to the relationship between data discovery and metadata richness.  The reviewers express this skepticism in slightly different ways, but I am in full agreement with them regarding this issue.   I also urge the authors to improve the writing and will be making a careful pass on the revision before deciding whether this paper has reached a level where more peer review is worth the effort.

We would appreciate receiving your revised manuscript by Nov 14 2019 11:59PM. To enhance the reproducibility of your results, we recommend that if applicable you deposit your laboratory protocols in protocols.io, where a protocol can be assigned its own identifier (DOI) such that it can be cited independently in the future. For instructions see: http://journals.plos.org/plosone/s/submission-guidelines#loc-laboratory-protocols

We look forward to receiving your revised manuscript.

Kind regards,

Robert Guralnick

Academic Editor

PLOS ONE

**Journal Requirements:**

**Comments to the Author**

1. Is the manuscript technically sound, and do the data support the conclusions?

Reviewer #1: Partly

Reviewer #2: Yes

2. Has the statistical analysis been performed appropriately and rigorously? 

Reviewer #1: Yes

Reviewer #2: Yes

3. Have the authors made all data underlying the findings in their manuscript fully available?

Reviewer #1: Yes

Reviewer #2: Yes

4. Is the manuscript presented in an intelligible fashion and written in standard English?

Reviewer #1: No

Reviewer #2: No

5. Review Comments to the Author

Reviewer #1: This paper addresses the suitability of metadata standards and metadata fields in 5-6 repositories to support searches for biodiversity-related research. My understanding is that search interests of biodiversity researchers were evaluated using 169 relevant research questions generated by 73 scientists, subsequently annotated. And by evaluating metadata documents in six repositories, including generalist repositories, and Pangaea, GBIF. I like the premise of evaluating dataset search needs based on common domain-specific research questions and comparing to metadata standards and existing metadata documents. However, it is very difficult for generalist repositories to effectively have domain-specific metadata requirements. This may be beyond the scope of this paper, but it also seems that you would need to evaluate content of metadata provided in general fields as well to understand findability and whether metadata are rich. In general, I find that this paper needs a lot of work to improve and clarify the writing, organization content, and methods used. It may help to first think about cutting the length significantly and focusing on key points and what you actually did.

Here are some other specific suggestions:

I suggest changing the title, as it is not quite grammatically correct, and “findable” is of course only one component of FAIR. Having FAIR in the title implies that you would some way address all aspects of FAIR.

The first paragraph of the abstract discusses data reuse in general, but this is not what the paper is about. This paragraph should be revised to be specifically relevant to the paper. I suggest in this revision to ensure that the first sentence is eye catching and relevant. “For a number of reasons,” is not a great start for an abstract. The abstract in general is too vague. You should briefly indicate the general method for determining search interests here.

There are grammatical errors throughout the paper, and it should be generally revised for grammatical correctness. There are too many errors to list here.

DataONE is working on quantifying FAIR in datasets and in repositories. They have found that Findability and Accessibility are covered reasonably well, but interoperability and reusability are not at all. You might look into what they are doing. At least address the fact that while Findability is still an issue, we are doing better here than in other aspects of FAIR.

See: https://www.dataone.org/webinars/quantifying-fair-metadata-improvement-and-guidance-dataone-repository-network

Specific checks they have developed for Findability are described here: https://github.com/NCEAS/metadig-checks/labels/Findable

You have to consider the scope of a particular repository to determine what is feasible to use as a metadata standard. A generalist repository cannot serve specific search interests for each domain represented. Therefore much of the responsibility of making data findable rests on the scientists describing their data. There are also opportunities for this are in general fields: the title, abstract, keywords, and methods, where they can ensure that relevant terms are included.

Line 77: Others are working on evaluating FAIR, and Findability happens to be the most well-covered thus far. There is definitely a lot of work to be done here, but is this statement is accurate?

Authors refer to “biodiversity” as a field. This should be “biodiversity research/science” or “biodiversity informatics.” And, biodiversity data is much further along in achieving FAIR than most other domains, such as earth and environmental science. Bio-medical data is the farthest along here, as you are aware, but other fields are behind biodiversity science.

Authors need to define better what you mean by “biodiversity” field and the kind of information you are looking for in the different repositories early on in the abstract and introduction.

Much of the Related Work section does not seem focused on the research questions (which are also unclear as mentioned previously). The paper should focus more on what the authors actually did, and less on specific methods done in past studies. For example, you should discuss how search works in the specific repositories where you evaluate metadata.

169 questions generated by 73 scientists does not seem like a sufficient dataset for broad search needs in biodiversity research. Although, the general categories of information produced seem reasonable.

In recommendations to data repositories in Table 10: What do you mean by #4? For most repositories, data providers are responsible for “filling in” metadata elements and providing descriptive metadata. And, it is not possible to do this when serving a broad community.

In evaluating metadata usage in repositories and findability, I do not understand how you determined whether metadata provided in the repositories was rich? Is it only by whether certain fields have associated metadata provided? You can have rich metadata descriptions in general fields, title/abstract/methods with terms that address the topic areas you found to be important. Particularly for generalist repositories, you cannot have an abundance of domain-specific metadata elements. To evaluate metadata richness for findability, I think that a better approach would be to look at the actual terms provided in the general metadata elements (e.g. title/keywords/abstract). The content analysis mentioned for specific repositories seems somewhat haphazard. Did you have a specific approach for this?

Lines 1279-1284: Many fields may not be relevant for a specific dataset, so filling in all metadata fields is not a realistic recommendation.

Reviewer #2: This is a study about the completeness of metadata in scientific repositories within the biodiversity realm. Authors argue that biodiversity datasets can only be found if rich metadata are available for the sets, and build a test using an annotated search term corpus through surveys within specific communities that they later use to characterize metadata schemata in common repositories and to group metadata terms into basic categories. Armed with this framework, they set to verify the metadata completeness of a sample of repositories by parsing the metadata of a large number of datasets within each repository, looking at what fraction of each relevant metadata field is filled.

The results describe the relative completeness of metadata across repositories, and therefore the relative completeness of the metadata categories. Overall, they aim to provide a picture of the findability of datasets through the probability of matching search terms to metadata contents.

The paper partially succeeds at this. The main objection is that the paper is constructed around the idea of the direct correspondence between findability and metadata richness. While this correspondence exists, it is not the only FAIR factor in the specific case of biodiversity research.

The introduction thoroughly describes the common methodology for metadata analysis applied to the general case—but in the specific case of biodiversity, a significant amount of searching occurs directly on the repository contents. A paramount example is one of the analyzed repositories: GBIF. Searches in GBIF are done through structured querying within the index and therefore can easily bypass the metadata descriptors (unless one considers all DwC-compliant data in the index as metadata!). Therefore, the methods described are less significant to gauge the retrievability from the common search than is for other types of repositories. The numerous considerations and recommendations for filling metadata might therefore become moot if the search object is the data contents rather than the metadata contents.

Consequently, while the majority of the results hold for the general case or findability in non-specific repositories, the description of the study and its results seem confusing or shallow when applied to the specific case of retrievability of biodiversity data. A clearer examination of the retrieval mechanisms; a discussion of the particularities of biodiversity data repositories, and a comparison to other completeness studies, seem necessary.

Formally, the paper needs some reworking as well. There are quite a number of language issues that should be addressed. I suggest a full pass by a domain-familiar native speaker. Although I offer some suggestions below, these are by no means complete and should only be taken as examples of the type of rewriting that needs to be done.

Some plots and graphs are overly atomized and can be condensed. For example, figures 3a and 3b can be reduced to one single figure with two series. The paper could certainly benefit from quite some simplification, concentrating on the main results and recommendations.

Below I list a number of specific observations and issues referenced by line number.

40 “Raw data described with”

81 I believe this is “dataset retrieval”. A dataset search can hardly be characterized as an application—it is a procedure, that may use an application.

113-114 There is literature already available about user needs in the realm of biodiversity data.

123 This definition of metadata richness seems a bit restrictive. Metadata describe the data, and therefore can be rich if the description is as complete as possible. This completeness helps ensuring search success, but search is not the only reason for metadata. Rich metadata may also help classifying datasets or deriving knowledge from the metadata, rather than from the metadata they represent.

156 I disagree in that a Research Question must be broad and semantically complex. There are research questions which may be formulated in very narrow terms and perhaps be directly answered by simple searches. For example, this very paper addresses whether metadata are rich enough for efficient search in biodiversity. An associated research question would be what the user needs are when they query biodiversity databases: in fact, an answer to this question is required to properly formulate the other question, for searches in biodiversity are driven by what the users’ needs for information or knowledge are. That research question (user needs) might, and has been, solved by a meta-analysis of literature where the data used were in fact literature metadata using a single query (Ariño et al., 2018: DOI:10.3897/biss.2.25738) where keywords are tallied, and by a survey (Faith et al., 2013: DOI:10.17161/bi.v8i2.4126) using highly focused questions addressed to a dataset of practitioners. While the authors may be tempted to dismiss these as “search questions”, they are also final goals in themselves (thence, “research questions”). In my view, the distinction that authors make between Research Questions and Search Questions is not correctly formulated. It seems as if the authors actually wanted to make the distinction as having metadata as a gateway to answer research questions, while search questions would be answered by the data described by the metadata (but not by the metadata), but not explicitly stated as such. At any rate, I don’t think the distinction applies.

186 “a list of terms and their frequencies—how often they…”

194 “are returned first”

p. 46, Fig. 1: the color scheme is not clear at first sight. Intended to mean: blue, backend: brown: user interface. Shadowing the BACKEND and USER INTERFACE titles would help. Footer (p. 6) says Green for crawling but no green is visible in the figure.

235 A slightly expanded description of what a TREC is, including expansion of the acronym, may be convenient here in addition to the reference.

247 a gold standard for what? For PR?

251 phrase unclear, does it mean “the TopN-ranked documents to compute…”?

257 phrase unclear – higher scores or scores higher?

258 what is “space” in this context? Also, use plural.

261 analysis of the underlying

263 either “sources of data […] are” or “source of data […] is”

270- The RDA DDG study mentioned here concerns itself with general search systems but this paper focus on biodiversity data repositories. The community of users tends to use a fundamentally different approach where the search granularity afforded by specific engines grants precise resultsets based not on keywords but on actual contents. The argument that “there is a need to analyze the given metadata and to quantify the gap between a scholar’s actual search interests and the underling metadata” is valid for the general case, but perhaps less so for the specific case of biodiversity data. A deeper and more tailored argument, with examples, might be required here.

357 I don’t know about such question corpus aimed at metadata, but I also very much doubt that a corpus of questions aimed at retrieving specific datasets does not exist. For example, GBIF stores al queries resulting in resultsets and caches them for later, more expedite answer if the question is repeated and the underlying data have not changed. This, by definition, should constitute a question corpus—but I do not know whether the corpus, as such, is public or available on demand. I believe that the statement in lines 357/358 should more properly phrased as “there is no published log analysis of a question corpus for the biodiversity domain”.

359-361 Or, alternatively, the actual user focus IS well served by the existing datasets!

391 “…on subject, predicate and object triples.”

397 “The result set contains…”

426 – 428 rephrase

466-477 The summary of the methodology is rather unclear and remains so until the detailed description has been read. Either write (or, better, plot) a clearer summary, or do away with it.

623 “all of them have experience…”

679 “plant residal” – is this verbatim or was intended “plant residual”?

683 “decreased”

703 “inter-rated”

710 “paradox”

732 “This is consistent with…”

744 “One-term artifacts”

768 “improvement”

776 Don’t these low thresholds reduce reliability for this study?

782 “shortened”

783 “insights in what scholars…”

785 “gets lost”

791 “selected”, “explored”

Table 2 Darwin Core is now named Darwn Terms.

825 “Dublin Core. We suppose…”

836 “and therefore is the only…”

848 “we decided not to use it.”

856 “Our results are”

861 I presume this is yellow?

864 “This was not expected.”

895-898 “Although there are … (geospatial data), important search topics…”

903 “should consider adding…”

907 Isn’t it six repositories, at least initially?

926 “The results are”

949 “in this list. OAI-DC…”

956 “to more fine-grained..”

1025 “a fixed number”

[Figures 3a&b could be combined. Figures 4a&b could be combined. Figures 5a&b could be combined. Why red bars?

Figure 7b, 8b, 9b, 10b, 11b: Apparently the Y axis is not percent but fraction. Difficult to read. Suggest vertical format. What is the ordination criterion? Why not ordered by %?

Figure 7a, 8a, 9a, 10a, 11a: why not cumulative? Trends are much more visible and easier to compare. Also, data for 2019 is irrelevant: the year is not over yet.

1103 Unclear what information is added when: the information is added at the of collection, or the date of collection is the information that is added?

1103 Unclear what are the main important topics

1125 The statement is highly debatable. Search engines tailored to contents (such as e.g. GBIF) are very highly specific (albeit also highly structured) and often do not require metadata search at all in order to produce and retrieve resultsets.

1126-1129 please rephrase and place punctuation correctly.

1136 “and analyzed them manually in a first round.”

1138 “as the main information needs”

1143 Genre means type here?

1147-1148 “We analyzed whether the elements of the metadata schemata reflected the identified information categories.”

1182-1194 needs rewriting. Also, the main idea is incomplete. As I said before, certain repositories e.g. GBIF are aimed at producing resultsets from queries redirected to their contents and can bypass metadata search entirely. It is difficult to apply the results of metadata completeness to such cases. Data completeness has been studied and published; for example, Gaiji et al. (2013) (DOI: 10.17161/bi.v8i2.4124) and Otegui et al. (2013) (DOI: 10.1371/journal.pone.0055144) in the case of GBIF. It would be highly desirable to discuss this difference and explore the interplay between these two approaches.

Table 10 omits the case of content-findable repositories.

1233 “that have not been considered so far are ”

1235 “and whether a controlled”

1250 bioschemas.org, not bioschema.org

1253 “At the time of writing”

1281 “As this is an emerging procedure”

1287 “whether the data is accessible”

1303 “Another challenge lies in the automatic”

1316 “independently of whether”

1326-1327 Unclear whether search interests cannot be found in existing metadata at all, or some search interests are not findable. Please clarify.

6. PLOS authors have the option to publish the peer review history of their article (what does this mean?). If published, this will include your full peer review and any attached files.

Reviewer #1: Yes: Joan E. Damerow

Reviewer #2: No

---

## [Author Response · Author response to Decision Letter 0]

5 Dec 2019

Dear Reviewers, 

We thank you for your kind comments and for your thorough feedback on our manuscript following our submission for publication. In accordance to the addressed issues, we have revised the manuscript. 

Please find attached to this letter a detailed reply on all issues (Response to Reviewers). All line numbers refer to the manuscript with track changes.

We look forward to hearing from you regarding our submission. We would be glad to respond to any further questions and comments that you may have.

Kind regards,

Felicitas Löffler

PhD student and Research Associate

Friedrich Schiller University Jena, Germany

---

## [Decision Letter · Decision Letter 1]

2 Apr 2020

PONE-D-19-18843R1

Dataset search in biodiversity research: Do metadata in data repositories reflect scholarly information needs?

PLOS ONE

Dear Mrs. Löffler,

Thank you for submitting your manuscript to PLOS ONE. After careful consideration, we feel that it has merit but does not fully meet PLOS ONE’s publication criteria as it currently stands. Therefore, we invite you to submit a revised version of the manuscript that addresses the points raised during the review process.

The reviewers were of the opinion that this article is worthy of publication and has substantially addressed the concerns raised on the previous version.  There are, however, numerous requests for corrections and clarifications as enumerated by the reviewers.  These need to be addressed.  Reviewers also comment on the relation to FAIR and the recommendations made - these should be considered carefully.  One reviewer provides many structural recommendations that may improve the article in various ways; while these do not materially affect the content, any changes that improve readability should be considered.

We would appreciate receiving your revised manuscript by May 17 2020 11:59PM. To enhance the reproducibility of your results, we recommend that if applicable you deposit your laboratory protocols in protocols.io, where a protocol can be assigned its own identifier (DOI) such that it can be cited independently in the future. For instructions see: http://journals.plos.org/plosone/s/submission-guidelines#loc-laboratory-protocols

We look forward to receiving your revised manuscript.

Kind regards,

Hussein Suleman, PhD

Academic Editor

PLOS ONE

Reviewers' comments:

Reviewer's Responses to Questions

**Comments to the Author**

1. If the authors have adequately addressed your comments raised in a previous round of review and you feel that this manuscript is now acceptable for publication, you may indicate that here to bypass the “Comments to the Author” section, enter your conflict of interest statement in the “Confidential to Editor” section, and submit your "Accept" recommendation.

Reviewer #1: (No Response)

Reviewer #2: All comments have been addressed

2. Is the manuscript technically sound, and do the data support the conclusions?

Reviewer #1: Yes

Reviewer #2: Yes

3. Has the statistical analysis been performed appropriately and rigorously? 

Reviewer #1: Yes

Reviewer #2: Yes

4. Have the authors made all data underlying the findings in their manuscript fully available?

Reviewer #1: Yes

Reviewer #2: Yes

5. Is the manuscript presented in an intelligible fashion and written in standard English?

Reviewer #1: Yes

Reviewer #2: Yes

6. Review Comments to the Author

Reviewer #1: This manuscript provides an important contribution, analyzing search needs derived from research questions when conducting dataset searches for biodiversity research. The authors compare concepts identified in the actual research questions to metadata elements from existing standards and metadata usage metrics in repositories. The paper is significantly improved and I think should be accepted, but still requires major revisions. In general the organization still needs improvement, and the background information can be cut and some components integrated into the discussion to place results in the context of previous work. The other major area for improvement is to better incorporate some specific results from your content analysis of search questions into the recommendations. Here are some more specific comments:

Abstract

- Last sentence: remove “how”, change to “recommendations for researchers and data repositories to bridge the gap…”

- Lines 13-15: change to “…reports that 40% of users attempting to search within two open data portals could not find the data they were interested in, and thus directly requested it from the repository manager.”

- Line 39: change to “We first identified main entity types…”

- Lines 50 - 52: Sentence starting with “In dataset search, he main sources…” is a bit unclear to me. DO you mean “The main options for search include metadata…”?

- Lines 52-53: change to: “The metadata schema required or recommended by the repository greatly influences the richness of metadata descriptions and determines available facets for filtering.”

- Line 63: change to “… and determined whether they utilize metadata elements that reflect search interests.”

- Line 126: focusing misspelled

- Line 136: this is unclear, please clarify this sentence starting with “Information needs can range from specific questions…”

- Line 150: The intro to background is a bit unclear. Do you mean: “This section provides background information on a variety of inputs involved in a search process…”?

- Is detailed background on the retrieval process necessary? I think that brief information on the exact retrieval process used in the repositories analyzed would be more useful for this paper. And focus on the situation with dataset retrieval and searches focused on standard metadata.

- There is also too much detail in the related work section. You can greatly reduce the amount of background information provided by focusing specifically on background information directly related to this work. I suggest some brief statement (a couple of sentences or maybe paragraph) about work in other fields followed by specific information that is more directly relevant. The GBIF portion is most relevant. The interesting thing here is that we can use methods developed and more thoroughly explored in other fields to answer questions related to biodiversity dataset publication and use. You do not need the details of results for those other studies.

- Line 275: change “scholars used retrieved…” to “scholars retrieved and used…”

- The objectives section is redundant with the objectives already presented in introduction? I think you can cut out multiple pages here still. Maybe even completely cut or move to supplement the entire Retrieval Process, Related Work, Dataset Search, Objectives section. You could summarize all of this into a couple of paragraphs in the main intro.

- Line 427-28: change to “…common sources to determine user interests for a particular domain.”

- Line 432: This is not an informative sentence and should be replaced

- Line 448: replace “which” with “that”

- Line 465: Lose spelled incorrectly as “loose”

- Line 470: why is EASE identified here but not discussed in the standards section? I am not familiar with this.

- Line 478: I do not see a good description of the Quality category. Do you have a better description of this that you provided to annotators?

- Line 502: replace “an own” with “a new”

- Figures: The titles for axes and legend items should be clarified and standardized across figures to improve presentation and clarity. Figures 6b-e have numerous items in legend, but for most of these it is not clear that all of these are present in the actual figure as presented

- Lines 537-539: The results section includes a mix of methods and results, so consider renaming this section.

- For the annotations of entities from the researcher questions, how do you map the entities/concepts that you identified to metadata fields from existing standards?

- Line 652: Change to: “However, such methods may result in biased feedback.

- Line 710: change to: “…to determine whether search interests can be explicity described with metadata elements from existing standards.”

- Lines 744-745: This is unclear, rephrase

- Table 4 title: The categories are sorted by frequency of what? Clarify the title

- Line 748: Change to “…they ask scholars to publish data with a repository for their research domain.” Ideally we want to focus on publishing data as an important contribution, not simply uploading files to a repository with no review process

- Line 838: Clarify what repository you are referring to for “The code and all charts are available in the repository.”

- Table 8 title: Again, frequency is not explained in the title.

- Line 882: change “Top5” to “five most common”?

- Line 886: Change the sentence starting with “None of the repositories…” to something like: “There are inconsistencies in how terms are entered; for example,

- In general, the metadata usage section I think is really important, but the writing is a little hard to follow and not as well organized. You need to reorganize this section to clarify key points. I really like how you are identifying what metadata elements are identified in general fields like keywords, versus more specific metadata elements, but this is not well explained and hard to follow.

- Line 899: change “which kind of entity occur” to “entity types that occur”

- Line 904: Why would you filter using a group of relevant keywords for Dryad, and only “Biodiversity” for Zenodo and Figshare? This does not make sense to me.

- Figshare has a bias towards collection data?

- Lines 940-941: I think you should remove or revise this. It is a real problem if the comprehensibility of your categories is only “fair”. However, Quality seems like the only one that is not well defined and less intuitive. The others match fairly well with metadata elements from existing standards. I also think that you need to better define these categories up front.

- Line 945: change “Elements of general standards cover the categories to some extent, only” to “…to some extent.”

- The discussion should usually present results in the context of previous work, but currently only summarizing your results. You need to revise the discussion to concisely present your results in the context of other work done in this arena. Maybe taking some of the information presented in the background section. However, caution to balance providing some useful and specific context without going into too much detail (as is the case currently in the background section).

- Line 969: remove “We figured that” from the sentence

- Lines 970-971: What do you mean by this - “repositories did not fully use all provided elements.” And “Most repositories seem to be aware of that problem and enhance metadata with numerous keywords in generic fields such as dc:subject.” The responsibility is usually on the dataset authors to provide this information. Maybe more detail should be required by the repository, and of course it would be nice if there were more of a review process for datasets, but the resources/incentives are not currently there. So, unless I misunderstand the meaning, the current language is not accurate.

- Line 975: clarify sentence, what supports improved filtering in search?

- Table 12 Recommendations: I don’t understand how #4 is for repositories. Should this not be for authors? Repositories are not responsible for describing the data, but they are responsible for providing good user interfaces and options for controlled vocabularies and standards. Also, these recommendations are too general in some cases and should incorporate some of the specific metadata recommendations identified to improve findability for biodiversity research questions.

- Line 1002: typo with extra comma and space

- Line 1084: remove this sentence, doesn’t make full sense. You might at least mention incentives for scholars to provide rich metadata (there are a lot of related issues around publishers not reporting data citations, data citations metrics behind paywalls, not sufficient resources for repositories to review datasets thoroughly etc.).

Reviewer #2: This amply revised version is much improved and satisfactorily addresses the concerns and issues I raised on the previous version. In particular, amendments made to the methodological descriptions now enable a good understanding of the entire workflow and the retrieval and annotation processes. It is therefore easier to understand and enables replication. The writing style is still a little verbose but that does not hamper the readability; the manuscript has now no outstanding issue requiring a rewrite. The tables and figures are quite helpful at summarizing the results and provide a good overview of the full study.

Therefore I think I can be published now, although a few minor details arising in the revised manuscript still need to be corrected as listed below.

61 GBIF contains taxonomic data, but it is primarily an occurrence data repository.

123 cursive (“What”)

126 focusing

271 … research has been conducted…

340 … a variety of approaches has emerged …

345 occurring

401 We argue that in dataset search…

496 As a geological era, Triassic should be capitalized.

527 The latter also applied if they …

582 and following – The parameter is called “Gwet’s AC”, not GWET’s AC (proposed by Kilem Li Gwet in 2008). The capitalized thing is just the function name in some packages (GWET_AC).

892 occurred

7. PLOS authors have the option to publish the peer review history of their article (what does this mean?). If published, this will include your full peer review and any attached files.

Reviewer #1: Yes: Joan E Ball-Damerow

Reviewer #2: Yes: Arturo H. Ariño

---

## [Author Response · Author response to Decision Letter 1]

15 May 2020

Dear Reviewers,

We thank you for your thorough and constructive feedback on our manuscript following our submission for publication.

In accordance to the addressed issues, we have revised the manuscript. Please find attached to this submission a detailed reply on all issues. All line numbers in the right column refer to the manuscript with track changes. To facilitate the review, in the manuscript with track changes, we highlighted new sentences in red color. Revised paragraphs or paragraphs moved and revised are highlighted in blue color. Paragraphs that were moved without any change in content are highlighted in green.

We look forward to hearing from you regarding our submission. We would be glad to respond to any further questions and comments that you may have.

Kind regards,

Felicitas Löffler

---

## [Decision Letter · Decision Letter 2]

29 Jul 2020

PONE-D-19-18843R2

Dataset search in biodiversity research: Do metadata in data repositories reflect scholarly information needs?

PLOS ONE

Dear Dr. Löffler,

Thank you for submitting your manuscript to PLOS ONE. After careful consideration, we feel that it has merit but does not fully meet PLOS ONE’s publication criteria as it currently stands. Therefore, we invite you to submit a revised version of the manuscript that addresses the points raised during the review process.

This updated manuscript has addressed many issues noted by reviewers previously.  Additional useful comments have been made about the use of terms related to metadata and how the contributions are presented by the authors.  These comments are from the perspective of the specific research community but also link back to general concepts and current ideas in metadata, and need to be addressed.  A key comment is about the sampling of standards assessed by the authors - this is critical and needs to be discussed or revisited in the article.  All reviewers over the course of this review process have made positive comments about the potential contribution and usefulness of this article, so the comments from reviewers may seem substantial but they do not warrant a fundamental change in the article; what is required is some mostly justification and contextualisation.  However, as evidence-based research, if more sources do indeed result in changes in the conclusions, these should be made.

We look forward to receiving your revised manuscript.

Kind regards,

Hussein Suleman, PhD

Academic Editor

PLOS ONE

Reviewers' comments:

Reviewer's Responses to Questions

**Comments to the Author**

1. If the authors have adequately addressed your comments raised in a previous round of review and you feel that this manuscript is now acceptable for publication, you may indicate that here to bypass the “Comments to the Author” section, enter your conflict of interest statement in the “Confidential to Editor” section, and submit your "Accept" recommendation.

Reviewer #2: All comments have been addressed

Reviewer #3: All comments have been addressed

Reviewer #4: (No Response)

2. Is the manuscript technically sound, and do the data support the conclusions?

Reviewer #2: Yes

Reviewer #3: Yes

Reviewer #4: Partly

3. Has the statistical analysis been performed appropriately and rigorously? 

Reviewer #2: Yes

Reviewer #3: Yes

Reviewer #4: I Don't Know

4. Have the authors made all data underlying the findings in their manuscript fully available?

Reviewer #2: Yes

Reviewer #3: Yes

Reviewer #4: Yes

5. Is the manuscript presented in an intelligible fashion and written in standard English?

Reviewer #2: Yes

Reviewer #3: Yes

Reviewer #4: Yes

6. Review Comments to the Author

Reviewer #2: This third version of the manuscript builds on and improves on the previous two, and as all outstanding issues have been addressed is almost ready for publication. Only a few typos remain that should be corrected (listed below) plus a rather striking result that I want checked out.

Line 80: and and -> and

Table 8, caption: final full stop.

Line 853: Remove space before comma (Geographic information, e.g. …)

Table 9, fifth row, second column: Is it actually “Plantas” (in Spanish, as it appears on the table) or rather “Plants” or "Plant" (in English) or “Plantae” (the Latin term)? Please check.

Line 900: GFBIF’s -> GBIF’s

Line 915: reserach -> research

Line 954: Add space at the end (…data retrieval. Therefore,)

Line 1004: facililate -> facilitate

Line 924: author’s -> authors’

Reviewer #3: No additional revisions needed.

Authors have adequately addressed your comments raised in a previous round of review.

Reviewer #4: Please see attached comments

7. PLOS authors have the option to publish the peer review history of their article (what does this mean?). If published, this will include your full peer review and any attached files.

Reviewer #2: **Yes: **Arturo H. Ariño

Reviewer #3: No

Reviewer #4: No

---

## [Author Response · Author response to Decision Letter 2]

8 Oct 2020

Dear Reviewers,

We thank you for your kind comments and for your thorough and constructive feedback on our manuscript following our submission for publication.

In accordance to the addressed issues, we have revised the manuscript. Please find attached to this letter a detailed reply on all issues. All line numbers in the right column refer to the manuscript with track changes. To facilitate the review, we highlighted new sentences in red color. Revised paragraphs or paragraphs moved and revised are highlighted in blue color. 

We look forward to hearing from you regarding our submission. We would be glad to respond to any further questions and comments that you may have.

Felicitas Löffler

PhD student and Research Associate

Friedrich Schiller University Jena, Germany

---

## [Decision Letter · Decision Letter 3]

17 Nov 2020

PONE-D-19-18843R3

Dataset search in biodiversity research: Do metadata in data repositories reflect scholarly information needs?

PLOS ONE

Dear Dr. Löffler,

Thank you for submitting your manuscript to PLOS ONE. After careful consideration, we feel that it has merit but does not fully meet PLOS ONE’s publication criteria as it currently stands. Therefore, we invite you to submit a revised version of the manuscript that addresses the points raised during the review process.

The paper was reviewed primarily by the reviewer who recommended many changes on the previous version, and it was confirmed that the authors have substantially addressed the concerns raised.   The reviewer still has concerns about the portal used as a major data source, but these do not need to be addressed as it is clear (at least to the editor) why the article has merit based on the portal used, and why the portal enables the kinds of analysis that were conducted.  However, there were also other minor questions raised and some typographical errors that need to be addressed by the authors.

We look forward to receiving your revised manuscript.

Kind regards,

Hussein Suleman, PhD

Academic Editor

PLOS ONE

Reviewers' comments:

Reviewer's Responses to Questions

**Comments to the Author**

1. If the authors have adequately addressed your comments raised in a previous round of review and you feel that this manuscript is now acceptable for publication, you may indicate that here to bypass the “Comments to the Author” section, enter your conflict of interest statement in the “Confidential to Editor” section, and submit your "Accept" recommendation.

Reviewer #4: (No Response)

2. Is the manuscript technically sound, and do the data support the conclusions?

Reviewer #4: Partly

3. Has the statistical analysis been performed appropriately and rigorously? 

Reviewer #4: I Don't Know

4. Have the authors made all data underlying the findings in their manuscript fully available?

Reviewer #4: Yes

5. Is the manuscript presented in an intelligible fashion and written in standard English?

Reviewer #4: Yes

6. Review Comments to the Author

Reviewer #4: I have uploaded my review as an attachment.

7. PLOS authors have the option to publish the peer review history of their article (what does this mean?). If published, this will include your full peer review and any attached files.

Reviewer #4: No

---

## [Author Response · Author response to Decision Letter 3]

22 Dec 2020

Dear Reviewers,

We thank you for the comments and feedback on our manuscript following our submission for publication.

In accordance to the addressed issues, we have revised the manuscript. The issues addressed by the reviewers are contained in a separate file (Response to Reviewers). All line numbers in the right column refer to the manuscript with track changes. To facilitate the review, we highlighted new sentences in red color. Revised paragraphs or paragraphs moved and revised are highlighted in blue color. 

We look forward to hearing from you regarding our submission. We would be glad to respond to any further questions and comments that you may have.

Kind regards,

Felicitas Löffler

PhD student and Research Associate

Friedrich Schiller University Jena, Germany

---

## [Editor Report · Decision Letter 4]

14 Jan 2021

Dataset search in biodiversity research: Do metadata in data repositories reflect scholarly information needs?

PONE-D-19-18843R4

Dear Dr. Löffler,

We’re pleased to inform you that your manuscript has been judged scientifically suitable for publication and will be formally accepted for publication once it meets all outstanding technical requirements.

Kind regards,

Hussein Suleman, PhD

Academic Editor

PLOS ONE

Additional Editor Comments (optional):

Thank you for submitting your revised manuscript.  I reviewed your changes personally and you have appropriately addressed the minor issues that were present in the manuscript, which is all I required to be addressed from the previous version.
---

## [Editor Report · Acceptance letter]

2 Mar 2021

PONE-D-19-18843R4 

Dataset search in biodiversity research: Do metadata in data repositories reflect scholarly information needs? 

Dear Dr. Löffler:

I'm pleased to inform you that your manuscript has been deemed suitable for publication in PLOS ONE. Congratulations! Your manuscript is now with our production department. 

Kind regards, 

on behalf of

Dr. Hussein Suleman 

Academic Editor

PLOS ONE